# The palmitoyl acyltransferase ZDHHC14 controls Kv1-family potassium channel clustering at the axon initial segment

Shaun S Sanders[1][†]*, Luiselys M Hernandez[1], Heun Soh[2], Santi Karnam[1], Randall S Walikonis[2], Anastasios V Tzingounis[2], Gareth M Thomas[1,3]*

[1]Shriners Hospitals Pediatric Research Center, Lewis Katz School of Medicine at Temple University, Philadelphia, United States; [2]Department of Physiology and Neurobiology, University of Connecticut, Storrs, United States; [3]Department of Anatomy and Cell Biology, Lewis Katz School of Medicine at Temple University, Philadelphia, United States

*For correspondence:
ssande03@uoguelph.ca (SSS);
gareth.thomas@temple.edu
(GMT)

Present address: [†]Department of Molecular and Cellular Biology, University of Guelph, Guelph, Canada

Competing interests: The authors declare that no competing interests exist.

**Abstract** The palmitoyl acyltransferase (PAT) ZDHHC14 is highly expressed in the hippocampus and is the only PAT predicted to bind Type-I PDZ domain-containing proteins. However, ZDHHC14's neuronal roles are unknown. Here, we identify the PDZ domain-containing Membrane-associated Guanylate Kinase (MaGUK) PSD93 as a direct ZDHHC14 interactor and substrate. PSD93, but not other MaGUKs, localizes to the axon initial segment (AIS). Using lentiviral-mediated shRNA knockdown in rat hippocampal neurons, we find that ZDHHC14 controls palmitoylation and AIS clustering of PSD93 and also of Kv1 potassium channels, which directly bind PSD93. Neurodevelopmental expression of ZDHHC14 mirrors that of PSD93 and Kv1 channels and, consistent with ZDHHC14's importance for Kv1 channel clustering, loss of ZDHHC14 decreases outward currents and increases action potential firing in hippocampal neurons. To our knowledge, these findings identify the first neuronal roles and substrates for ZDHHC14 and reveal a previously unappreciated role for palmitoylation in control of neuronal excitability.

## Introduction

Neurons are large, morphologically complex cells whose normal physiological function requires targeting of different voltage- and ligand-gated ion channels to specific subcellular locations. For example, voltage-gated calcium channels and neurotransmitter receptors must be clustered pre- and post-synaptically, respectively, to ensure the fidelity of synaptic transmission (*Buonarati et al., 2019*; *Dolphin and Lee, 2020*; *Sanz-Clemente et al., 2013*; *Suh et al., 2018*). In addition, voltage-gated sodium and potassium channels must be clustered at the axon initial segment (AIS) and, in myelinated axons, at or near nodes of Ranvier, to ensure correct initiation and propagation of action potentials (*Catterall, 1981*; *Huang and Rasband, 2018*; *Leterrier, 2018*; *Lorincz and Nusser, 2008*; *Nelson and Jenkins, 2017*; *Waxman and Foster, 1980*).

Several mechanisms ensure the appropriate targeting of neuronal ion channels and receptors. In particular, non-enzymatic scaffold proteins, particularly those containing PSD95/discs large/ZO-1 (PDZ) domains, directly bind the intracellular C-terminal tails of several families of receptors and channels and thereby control their trafficking to, and/or stability at, precise subcellular sites (*Kim and Sheng, 2004*). PDZ domains are subdivided into several classes with the two best known subclasses, Type-I and Type-II PDZ domains, binding C-terminal sequences ending in S/T-X-V (where X is any amino acid) and Φ-X-Φ (where Φ is a hydrophobic amino acid), respectively (*Nourry et al., 2003*).

A second mechanism to control receptor, ion channel, and scaffold protein interactions and targeting is dynamic post-translational modification. One post-translational modification whose importance in neurons is increasingly appreciated is the covalent addition of long-chain fatty acids to protein cysteine residues, a process called S-palmitoylation (or S-acylation, referred to here simply as palmitoylation) (*Chamberlain and Shipston, 2015*; *Fukata and Fukata, 2010*; *Hallak et al., 1994*; *María-Eugenia and Van Der Goot, 2018*; *Smotrys and Linder, 2004*). Palmitoylation plays key roles in neurons (*Sanders et al., 2015*), particularly at synapses where many scaffold proteins and ion channels and receptors are palmitoylated (*Matt et al., 2019*; *Naumenko and Ponimaskin, 2018*; *Thomas and Huganir, 2013b*).

We previously suggested that these two mechanisms might act in concert because a surprising number of palmitoyl acyltransferases (PATs) possess C-terminal sequences that are themselves predicted to bind PDZ domain proteins (*Thomas and Hayashi, 2013a*). However, almost all of these potential PDZ-binding PATs terminate in Type-II PDZ ligand (Φ-X-Φ) sequences and hence are unlikely to directly bind major Type-I PDZ domain proteins, several of which are important palmitoyl-proteins in neurons (*Hung and Sheng, 2002*; *Kim and Sheng, 2004*). The only PAT that contains a C-terminal Type-I PDZ ligand (S/T-X-V) is ZDHHC14 (*Thomas and Hayashi, 2013a*), a PAT whose neuronal roles and substrates are unknown. The lack of investigation of ZDHHC14 is perhaps surprising, given that this PAT is highly expressed in the hippocampus (*Cembrowski et al., 2016*; *Zeisel et al., 2018*) and may even be the most highly expressed PAT in this brain region (*Cajigas et al., 2012*). ZDHHC14 is also one of only four PATs intolerant to loss of function genetic mutations in humans (*Lek et al., 2016*). This latter finding suggests that role(s) of ZDHHC14 cannot be compensated for by other PATs, which might be explained by ZDHHC14's unique predicted PDZ-binding ability.

Here, we identify the Type-I PDZ domain-containing Membrane-associated Guanylate Kinase (MaGUK) family scaffold protein PSD93 (post-synaptic density 93/chapsyn 110/DLG2) (*Kim et al., 1996*) as a direct interactor and substrate of ZDHHC14. Like its close paralog, PSD95, PSD93 is a known palmitoyl-protein (*El-Husseini et al., 2000b*; *Topinka and Bredt, 1998*). However, while palmitoylation is well described to control PSD95 targeting to the post-synaptic membrane (*Firestein et al., 2000*), the role of PSD93 palmitoylation is less clear. We provide evidence that palmitoylation by ZDHHC14 targets PSD93 not to synapses, but to the AIS. Loss of ZDHHC14 also reduces palmitoylation and AIS targeting of Kv1 family potassium channels, whose clustering at the AIS was previously reported to be PSD93-dependent (*Ogawa et al., 2008*). Consistent with this model, and with the known role of Kv1 channels in constraining action potential firing (*Yamada and Kuba, 2016*), loss of ZDHHC14 reduces outward currents, which are likely mediated by voltage-dependent potassium channels, and also increases neuronal excitability. These findings have implications for our understanding of physiological regulation of neuronal excitability and how such regulation is impaired in conditions linked to hyper-excitation.

## Results

### ZDHHC14 binds PSD93 in a PDZ ligand-dependent manner

Seven of 24 mouse and eight of 23 human PATs have sequences that terminate in a predicted PDZ ligand (*Thomas and Hayashi, 2013a*). However, only the sequence of ZDHHC14 terminates in a Type-I PDZ ligand (LSSV [Leu-Ser-Ser-Val-COOH]; *Figure 1A*). This LSSV sequence is perfectly conserved in vertebrates (*Figure 1—figure supplement 1A,B*) and might be predicted to bind Type-I PDZ-domain scaffold proteins (*Hung and Sheng, 2002*), many of which cluster ion channels and receptors at sites of excitability in neurons. We thus hypothesized that ZDHHC14 might use its conserved PDZ ligand to bind interacting partners and potential substrates and hence used this region as 'bait' to perform a yeast two-hybrid screen of a rat hippocampal cDNA library (*Dong et al., 1997*; *Thomas et al., 2012*; *Figure 1A*). Three unique 'hits' encompassed the third PDZ domain (PDZ3) of the scaffold protein PSD93 (*Figure 1—figure supplement 2A*). Back-transformation experiments using a shorter 'prey' plasmid expressing PSD93-PDZ3 alone revealed that PSD93-PDZ3 directly bound the wild type (LSSV) C-terminus of ZDHHC14 but did not bind a mutant ZDHHC14 C-terminus with a mutated PDZ ligand (LSSE: *Figure 1—figure supplement 2B*). These findings suggest that the C-terminal PDZ ligand of ZDHHC14 binds PSD93-PDZ3 in yeast.

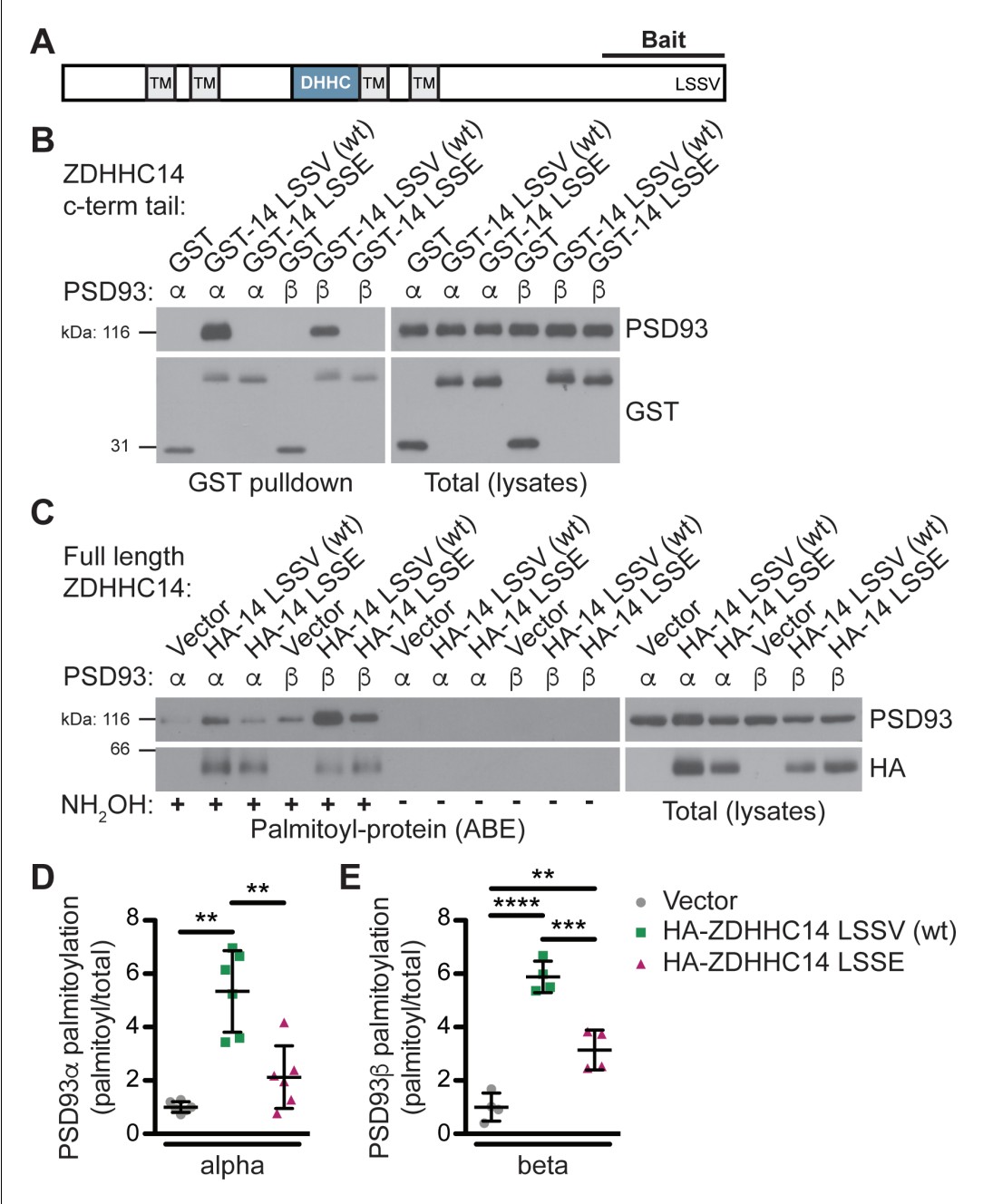

**Figure 1.** ZDHHC14 interacts with and palmitoylates both palmitoylated isoforms of PSD93 but more robustly palmitoylates PSD93β. (**A**) Schematic of ZDHHC14 showing predicted transmembrane domains (TM, gray boxes), DHHC cysteine rich catalytic domain (blue box) and the C-terminal region used for yeast 2-hybrid screening (Bait), including the LSSV motif. (**B**) HEK293T cells were transfected with the indicated constructs and lysates subjected to GST-pulldown. Eluates from pulldowns were immunoblotted to detect GST (bottom, left) and PSD93 (top, left). Total expression levels of GST-tagged proteins (bottom, right) and PSD93 (top, right) in parent lysates were also determined. Images are representative of three independent experiments. (**C**) HEK293T cells were transfected with the indicated constructs and palmitoyl-proteins (isolated by ABE; left panels) and total protein levels (in parent lysates; right panels) were assessed by western blotting with the indicated antibodies. Parallel samples processed in the absence of the key ABE reagent hydroxylamine (NH$_2$OH) confirm the specificity of the ABE assay. (**D**) Quantified PSD93α palmitoyl:total levels from C, normalized to the empty vector condition (Welch's 1-way ANOVA p=0.0008, W(2,6.98) = 23.80, N = 6; Dunnett's T3 multiple comparison *post hoc* test **p<0.01, 95% CI vector versus wtZDHHC14 [−6.91,–1.77], vector versus ZDHHC14 LSSE [−3.10, 0.86], and wtZDHHC14 versus ZDHHC14 LSSE [0.60, 5.83]). (**E**) Quantified PSD93β palmitoyl:total levels from *C*, normalized to the empty vector condition (1-way ANOVA p<0.0001, F(2,9)=60.69, N = 4; Bonferroni *post hoc* test **p<0.01, ***p<0.001, ****p<0.0001, 95% CI vector versus wtZDHHC14 [−6.18,–3.56], vector versus ZDHHC14 LSSE [−3.44,–0.84], and wtZDHHC14 versus ZDHHC14 LSSE [1.44, 4.04]). Uncropped western blot images are in *Figure 1—figure supplement 4*.

*Figure 1 continued on next page*

*Figure 1 continued*

The online version of this article includes the following source data and figure supplement(s) for figure 1:

**Source data 1.** Source data for *Figure 1D* and *Figure 1—figure supplement 3*.
**Figure supplement 1.** The PDZ ligand of ZDHHC14 is highly conserved in vertebrates.
**Figure supplement 2.** Further Yeast Two-Hybrid analysis suggests that the ZDHHC14 C-terminus binds the third PDZ domain of PSD93.
**Figure supplement 3.** Further analysis of palmitoylation:total levels of PSD93α and PSD93β with or without wt or LSSE ZDHHC14 from *Figure 1C*.
**Figure supplement 4.** Uncropped Western blot images for *Figure 1*.

We next sought to validate the interaction between PSD93 and ZDHHC14 in a mammalian system, using full-length PSD93. A glutathione-S-transferase (GST) fusion of the initial ZDHHC14 bait (GST-ZDHHC14-LSSV [wt], abbreviated as GST-14-LSSV (wt) on Figure) robustly bound each of two previously reported palmitoylated PSD93 isoforms, α and β (*Brenman et al., 1996*; *El-Husseini et al., 2000a*) in lysates of co-transfected HEK293T cells (*Figure 1B*). In contrast, neither PSD93 isoform bound a GST fusion of ZDHHC14 carrying a mutated PDZ ligand (GST-ZDHHC14-LSSE (GST-14-LSSE on figure); *Figure 1B*). ZDHHC14 and PSD93 are thus *bona fide* binding partners that interact in a PDZ-ligand-dependent manner.

## ZDHHC14 palmitoylates PSD93 in cotransfected cells

To determine if ZDHHC14 also palmitoylates PSD93, we co-expressed PSD93α or β in HEK293T cells with or without wild type HA-tagged ZDHHC14 (HA-ZDHHC14-LSSV [wt], HA-14-LSSV [wt] on Figure 1C). We then isolated palmitoyl-proteins using the acyl-biotin exchange (ABE) assay, a non-radioactive technique to purify palmitoylated proteins from cell lysates (*Roth et al., 2006*; *Thomas et al., 2012*). Palmitoylation of both PSD93α and β was low, though detectable, in the absence of cotransfected HA-ZDHHC14 (*Figure 1C,D,E*). However, HA-ZDHHC14 robustly increased palmitoylation of both PSD93 isoforms, with PSD93β palmitoylation increased to a much greater extent than that of PSD93α (*Figure 1C* [lane 2 versus 5], D, E, and *Figure 1—figure supplement 3*). These data suggest that ZDHHC14 can palmitoylate both PSD93α and β, but more robustly palmitoylates the β isoform. Interestingly, the HA-ZDHHC14-LSSE mutant (HA-14 LSSE on figure) palmitoylated both PSD93 isoforms less effectively than did wild type HA-ZDHHC14 (*Figure 1C* [lanes 2 versus 3 and 5 versus 6], D, E, and *Figure 1—figure supplement 3*). This finding suggests that the LSSV PDZ ligand is important, not only for ZDHHC14 to bind PSD93, but also to recognize it as a substrate.

## ZDHHC14 is the major neuronal PAT for PSD93

To determine if ZDHHC14 is the major PSD93 PAT in neurons, we transduced cultured rat hippocampal neurons with lentivirus expressing GFP with or without a short hairpin (sh) RNA targeting rat *Zdhhc14* mRNA (*Zdhhc14* sh#1) on day in vitro (DIV) 9. Robust knockdown of ZDHHC14 protein was achieved one week after infection (>90%; *Figure 2A,B*), at which point palmitoyl-proteins were isolated from control and *Zdhhc14* sh#1-transduced hippocampal neurons using the ABE assay. PSD93 was readily detected in ABE samples and its palmitoylation was decreased by >60% in *Zdhhc14* 'knockdown' cultures (*Figure 2C,D*). Total levels of PSD93 were also decreased after *Zdhhc14* knockdown, but to a far lesser extent than palmitoylation (*Figure 2C,E*). In contrast, palmitoylation and levels of another palmitoyl-protein, GAP43, were unaffected by *Zdhhc14* knockdown (*Figure 2A,C, D*). ERK, a non-palmitoylated protein, was not detected in ABE samples, confirming the specificity of the assay (*Figure 2C*). A second, independent *Zdhhc14* shRNA (*Zdhhc14* sh#2) also greatly decreased ZDHHC14 levels (*Figure 2—figure supplement 1A,B*) and PSD93 palmitoylation (*Figure 2—figure supplement 1C–E*), without affecting GAP43 palmitoylation (*Figure 2—figure supplements 1C, D*). These findings suggest that ZDHHC14 is the predominant PAT for PSD93 in hippocampal neurons.

## ZDHHC14 is required for targeting of PSD93 to the AIS

Although our results strongly suggested that ZDHHC14 controls PSD93 palmitoylation in neurons, this modification was reported to be dispensable for PSD93 targeting to synapses (*Firestein et al., 2000*). We therefore hypothesized that palmitoylation instead controls PSD93 targeting to a different subcellular location in neurons. Interestingly, PSD93 was previously reported to be the only

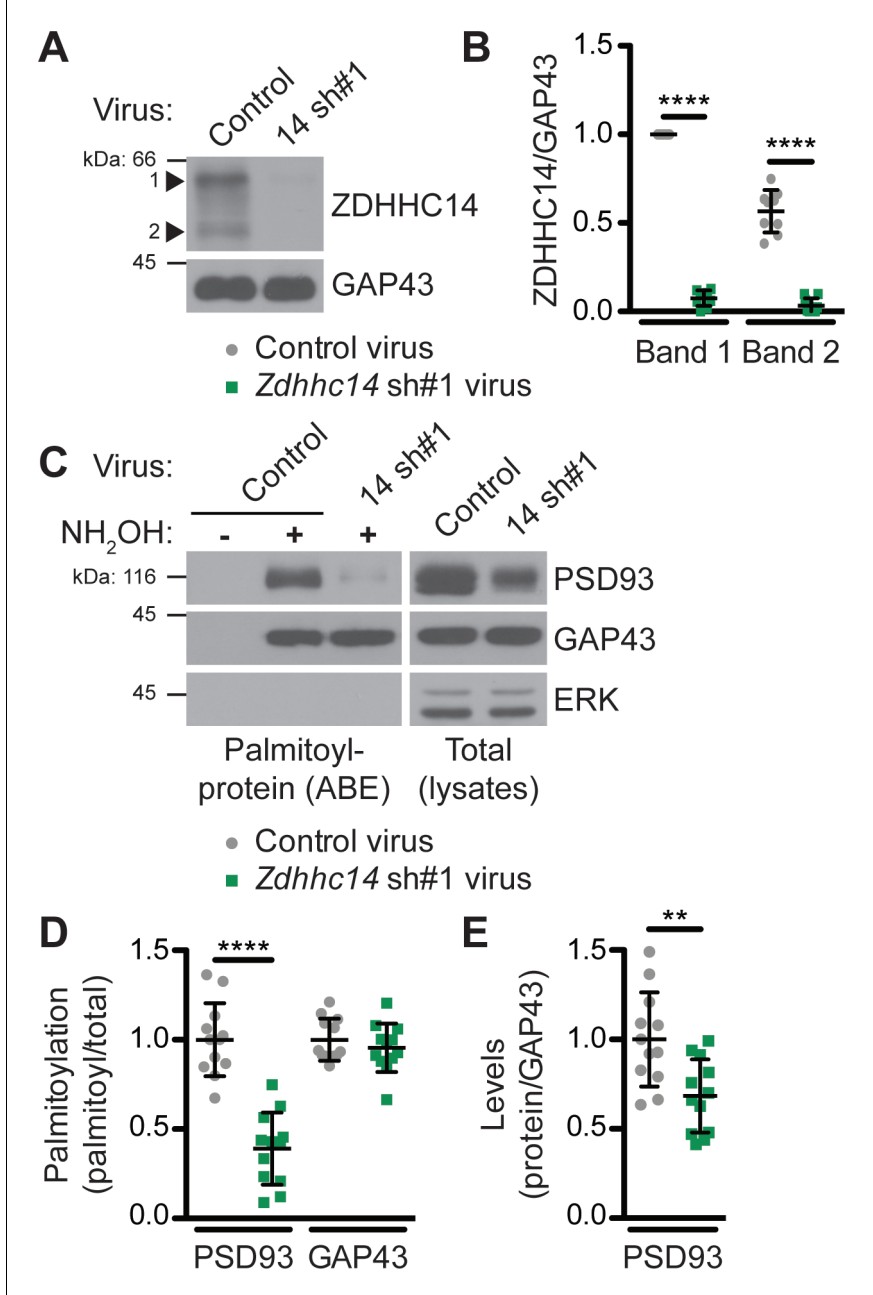

**Figure 2.** ZDHHC14 palmitoylates PSD93 in hippocampal neurons. (A) Cultured hippocampal neurons were transduced with the indicated lentiviruses on day in vitro (DIV) nine and lysed on DIV16. Lysates were blotted with the indicated antibodies. Two ZDHHC14 bands were identified (indicated by arrow heads '1' and '2'). (B) Quantified intensities of the indicated ZDHHC14 bands from *A*, normalized to band '1' in the control virus condition (2-way ANOVA: Virus p<0.0001 [F(1)=618.7], band p<0.0001 [F(1)=72.17], interaction p<0.0001 [F(1) =51.64]; N = 5; Bonferroni posthoc test ****p<0.0001; 95% CI control versus 14sh#1 band 1 [–1.01,–0.81], control versus 14sh#1 band 2 [–0.60,–0.40]). (C) Cultured hippocampal neurons were lentivirally infected and lysed as in *A* and palmitoyl-proteins (isolated by ABE; *left panels*) and total protein levels in parent lysates (*right panels*) were assessed by western blotting with antibodies against PSD93 (*top panel*), GAP43 (*middle panel*, positive control palmitoylated protein), and Erk1/2 (bottom, negative control non-palmitoylated protein). (D) Quantified data from *C*, showing PSD93 (left) and GAP43 (right) palmitoyl/total levels, normalized to the control virus condition (PSD93: unpaired Student's t-test ****p<0.0001, N = 12, 95% CI [0.44, 0.78]; GAP43: unpaired Student's t-test p=0.40, N = 12, 95% CI [−0.063, 0.15]). (E) Quantified data from *C,* showing total PSD93 levels normalized to the control

*Figure 2 continued on next page*

*Figure 2 continued*

virus condition (unpaired Student's t-test **p=0.0034, N = 12, 95% CI [0.12, 0.52]). Uncropped western blot images are in *Figure 2—figure supplement 2*.

The online version of this article includes the following source data and figure supplement(s) for figure 2:

**Source data 1.** Source data for *Figure 2B,D and E* and for *Figure 2—figure supplement 1B,D and E*.
**Figure supplement 1.** Additional Evidence that ZDHHC14 palmitoylates endogenous PSD93 in hippocampal neurons.
**Figure supplement 2.** Uncropped Western blot images for *Figure 2* and *Figure 2—figure supplement 1*.

---

MaGUK that also clusters at the AIS (*Ogawa et al., 2008*), so we asked whether palmitoylation might target PSD93 to this location. Consistent with prior findings, we robustly detected PSD93 at presumptive synapses (*Firestein et al., 2000*; *Kim et al., 1996*) and at the AIS (*Ogawa et al., 2008*; *Figure 3—figure supplement 1A*) in hippocampal neurons using standard fixation conditions. However, for our investigations of the importance of ZDHHC14, we used a previously described gentle fixation method, which preferentially reveals the AIS-localized pool of PSD93 (*Ogawa et al., 2008*; *Figure 3—figure supplement 1B*). Using this method, PSD93 was readily detected at the AIS, defined by enrichment of the master AIS scaffold protein Ankyrin G (AnkG or ANK3; *Figure 3A* and *Figure 3—figure supplement 1B*; *Huang and Rasband, 2018*; *Jenkins and Bennett, 2001*; *Kordeli et al., 1995*). However, PSD93 targeting to the AIS was significantly reduced in *Zdhhc14* knockdown neurons (*Figure 3B,C* [mean gray value]). Interestingly, AIS length was also reduced (by approximately 35%) in *Zdhhc14* knockdown neurons (*Figure 3D*). Taking the reduced AIS length into account revealed an even greater reduction of total AIS-localized PSD93 (integrated density) in *Zdhhc14* knockdown neurons (*Figure 3E*). *Zdhhc14* shRNA#2 also decreased PSD93 AIS targeting and AIS length (*Figure 3—figure supplement 2A–E*). Additional examples of reduced AIS targeting of PSD93 after *Zdhhc14* knockdown are shown in *Figure 3—figure supplement 3*. These findings are consistent with the hypothesis that palmitoylation by ZDHHC14 targets PSD93 to the AIS.

## Palmitoyl-site mutation reduces PSD93β AIS targeting

To more directly assess whether palmitoylation targets PSD93 to the AIS, we determined the extent to which mutation of PSD93's palmitoyl-cysteines reduces its AIS targeting. As ZDHHC14 more robustly palmitoylates PSD93β (*Figure 1C–E*), we sought to identify the site(s) on this isoform that are palmitoylated by ZDHHC14. There are five cysteine residues in the N-terminus of PSD93β: C10, C16, C18, C22, and C33 (*Figure 4—figure supplement 1A*). PSD93β was reported to be predominantly palmitoylated at C16 and C18, but this prior study *Firestein et al., 2000* used a truncated PSD93β cDNA lacking C10 and additional N-terminal sequences now ascribed to PSD93 (*Rattus norvegicus* XP_017445141.1, *Homo sapiens* NP_001338205.1). We found that ZDHHC14-dependent palmitoylation of full-length PSD93β-myc in which cysteines 16 and 18 were both mutated to non-palmitoylatable serine (C16,18S) was reduced by only 42% (*Figure 4—figure supplement 1B,C*), suggesting that ZDHHC14 palmitoylates additional sites on PSD93β. Consistent with this notion, mutation of C10 to serine (C10S) reduced ZDHHC14-dependent palmitoylation of PSD93β by 88% (*Figure 4—figure supplement 1B,C*). However, ZDHHC14–dependent palmitoylation was not completely blocked until all five N-terminal cysteine residues of PSD93β were mutated (*Figure 4—figure supplement 1B,C*; '5CS' mutant), suggesting that each of these sites can be palmitoylated by ZDHHC14.

To determine if direct palmitoylation of PSD93 is required for its AIS targeting, we compared the subcellular distribution of myc-tagged PSD93β wild type and 5CS mutant (wtPSD93β-myc or PSD93β−5CS-myc, respectively) in hippocampal neurons cotransfected with GFP as a morphology marker. Both wtPSD93β-myc and PSD93β−5CS-myc were targeted to punctate structures in dendrites (*Figure 4A,B*, respectively), which are likely synapses, in agreement with previous findings (*Firestein et al., 2000*). However, within the axon, wtPSD93β-myc was enriched at the AIS (*Figure 4A*) but PSD93β−5CS-myc was more diffusely distributed, extending past the AIS (*Figure 4B*). When PSD93β axonal distribution was scored as AIS-enriched or diffuse, wtPSD93β-myc was AIS-enriched in 67+/- 5.77% (SD) of transfected neurons whereas PSD93β−5CS-myc was AIS-

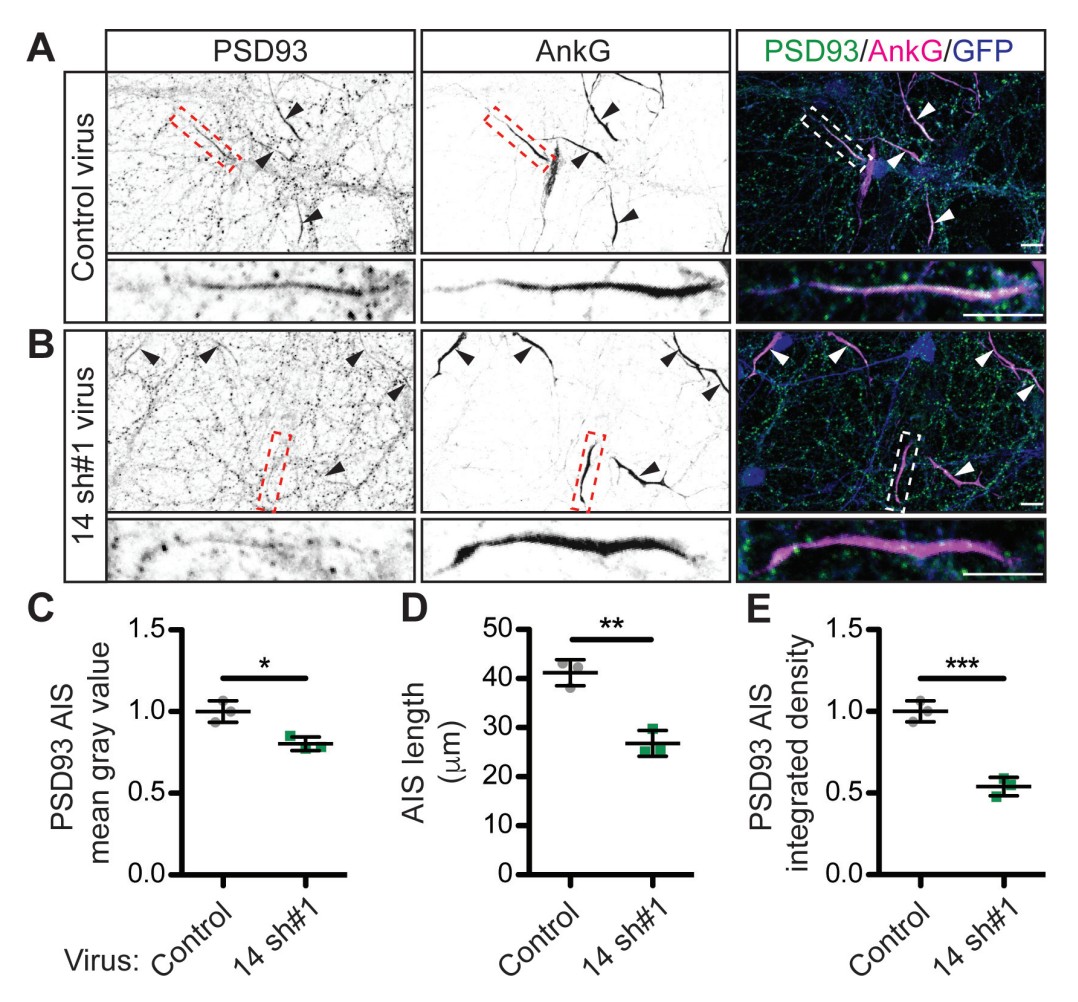

**Figure 3.** ZDHHC14 is required for PSD93 targeting to the AIS in hippocampal neurons. Hippocampal neurons transduced with lentivirus to express GFP alone (A; 'Control'), or together with *Zdhhc14* shRNA #1 (B; '14 sh#1') were fixed and immunostained with antibodies against PSD93 (*left column* and green in merged images on right), AnkG (AIS marker, *middle column* and magenta in merged images on right), and GFP (marker of infected neurons, blue in merged images, *right column*). Lower panel images are magnified views of dashed red or white boxed area of top row images and arrowheads indicate additional non-magnified AISs in image. (C) Mean gray value of PSD93 signal within AnkG-defined AIS, normalized to control condition (unpaired Student's t-test ****p=0.012, N = 3 independent cultures, 95% CI [0.072, 0.32]). (D) Quantified AIS lengths (defined by AnkG staining; unpaired Student's t-test **p=0.0026, N = 3 independent cultures, 95% CI [8.42, 20.39]). (E) Integrated density of PSD93 signal within AnkG-defined AIS, normalized to control condition (Student's t-test ***p=0.0008, N = 3 independent cultures, 95% CI [0.32, 0.60]). Scale bar in full and magnified views: 10 μm. Additional example images are provided in *Figure 3—figure supplement 3*.

The online version of this article includes the following source data and figure supplement(s) for figure 3:

**Source data 1.** Source data for *Figure 3C–E* and for *Figure 3—figure supplement 2C–E*.

**Figure supplement 1.** A gentle fixation method highlights AIS-localized PSD93 in hippocampal neurons.

**Figure supplement 2.** Additional Evidence that ZDHHC14 is required for PSD93 targeting to the AIS in hippocampal neurons.

**Figure supplement 3.** Additional example images confirm that ZDHHC14 is required for PSD93 targeting to the AIS in hippocampal neurons.

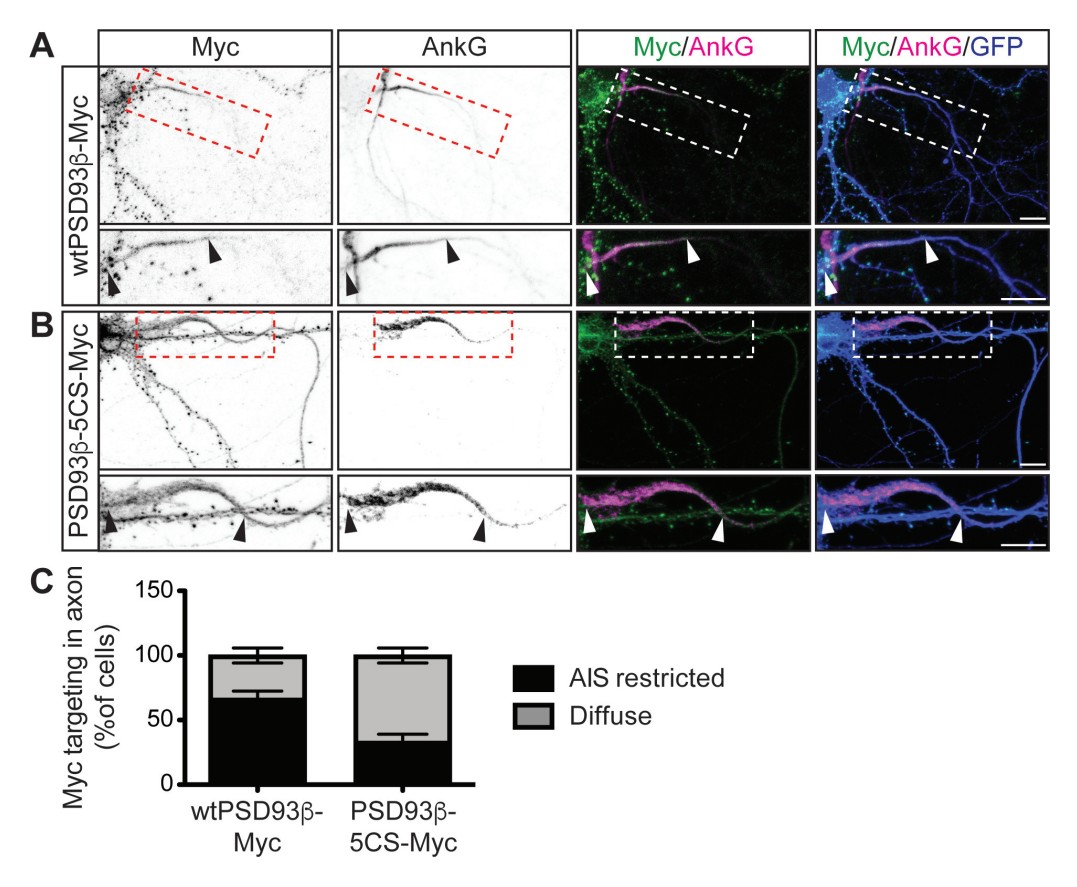

**Figure 4.** Contribution of direct palmitoylation to AIS targeting of PSD93β in hippocampal neurons. Hippocampal neurons were transfected to express GFP and wtPSD93β-Myc (**A**) or PSD93β−5CS-Myc (**B**). Neurons were fixed and immunostained with antibodies against Myc (*left column* and green in merged images), AnkG (AIS marker, *second column* and magenta in merged images), and GFP (marker of infected neurons, blue in merged images, *right column*). Magnified views of dashed red or white boxed areas of top row images are shown below and arrowheads indicate the start and end of the AIS. Neurons were scored for Myc distribution in the axon, delineated by the presence of an AnkG-positive AIS, as diffuse axonal distribution (Diffuse) or AIS enriched/restricted distribution (AIS-enriched). (**C**) Quantified data for axonal images for 10 neurons per condition per experiment transfected as in *A-B*, from three independent cultures expressed as a percentage per culture (N = 3 independent cultures). Scale bar in full and magnified views: 10 μm.

The online version of this article includes the following source data and figure supplement(s) for figure 4:

**Source data 1.** Source data for *Figure 4C* and for *Figure 4—figure supplement 1C*.
**Figure supplement 1.** ZDHHC14 palmitoylates PSD93β on all five N-terminal cysteine residues.
**Figure supplement 2.** Uncropped western blot images for *Figure 4—figure supplement 1*.

enriched in only 33%+/- 5.77% (SD) of transfected neurons (*Figure 4C*). These findings suggest that direct palmitoylation is important, although not essential, for PSD93β targeting to the AIS.

## Developmental expression of ZDHHC14 mirrors that of PSD93 and Kv1 channels

At the AIS, PSD93 acts as a scaffold to cluster Type-I voltage-gated potassium (Kv1) channels (*Ogawa et al., 2008*). The Kv1 channel subunits found at the AIS in hippocampal neurons are Kv1.1, Kv1.2, and Kv1.4 (*Kole et al., 2007*; *Ogawa et al., 2008*) and prior work from non-neuronal cells suggested that Kv1 clustering by PSD93 likely involves binding of Kv1.1/1.2/1.4 C-termini, each of which terminates in a Type-I PDZ ligand, to PSD93's second PDZ domain (*Kim et al., 1996*; *Kim et al., 1995*). Interestingly, Kv1.1 was reported to be palmitoylated when expressed in non-neuronal cells (*Gubitosi-Klug et al., 2005*). We realized that the region surrounding the Kv1.1 palmitoyl-cysteine mapped in this prior study is highly conserved in Kv1.1 orthologs, and a homologous

cysteine is present in both Kv1.2 and Kv1.4 (*Figure 5—figure supplement 1A,B*). These findings raised the possibility that Kv1.1, Kv1.2 and/or Kv1.4 might be endogenously palmitoylated in neurons, potentially by ZDHHC14 and, further, that such palmitoylation might control AIS targeting of these Kv1 channels. The decreased AIS length after loss of ZDHHC14 (*Figure 3D*) is consistent with this notion, because decreased AIS length is a well-known homeostatic response to increased neuronal excitability (*Dumitrescu et al., 2016*; *Grubb and Burrone, 2010*; *Grubb et al., 2011*), as might occur if these Kv1 channels were not properly targeted to the AIS.

As a first step to determine if ZDHHC14 also regulates Kv1 channels, we examined the developmental expression of ZDHHC14, PSD93, Kv1.1, Kv1.2, and Kv1.4 in hippocampal neurons. Strikingly, the developmental profiles of all five of these proteins were almost identical, with each first detectable at DIV8 and steadily increasing to DIV16 (*Figure 5A,B*). In contrast, expression of the GLUN2B subunit of the synaptic N-methyl-D-aspartate (NMDA) glutamate receptor was already detected at DIV4 and peaked at DIV12 (*Figure 5A,B*). This coordinated developmental upregulation of ZDHHC14 with PSD93 and Kv1 channels is consistent with a shared function in hippocampal neurons.

## ZDHHC14 is a major neuronal PAT for Kv1.1, Kv1.2, and Kv1.4

Although our biochemical developmental time course experiment did not differentiate between somal, synaptic, or AIS pools of ZDHHC14, PSD93, and Kv1 channels, it raised the possibility that ZDHHC14 might regulate palmitoylation and/or AIS targeting of not just PSD93, but also Kv1 channels. To investigate this possibility, we assessed Kv1 channel palmitoylation in the absence of ZDHHC14. Consistent with our hypothesis, *Zdhhc14* knockdown reduced palmitoylation of Kv1.1, Kv1.2, and Kv1.4 by 35, 31, and 65%, respectively (*Figure 6A,B*). *Zdhhc14* knockdown also reduced total levels of Kv1.2 and Kv1.4, but to a lesser extent than palmitoylation (*Figure 6A,C*). These findings suggest that ZDHHC14 is a PAT for Kv1 channels in hippocampal neurons.

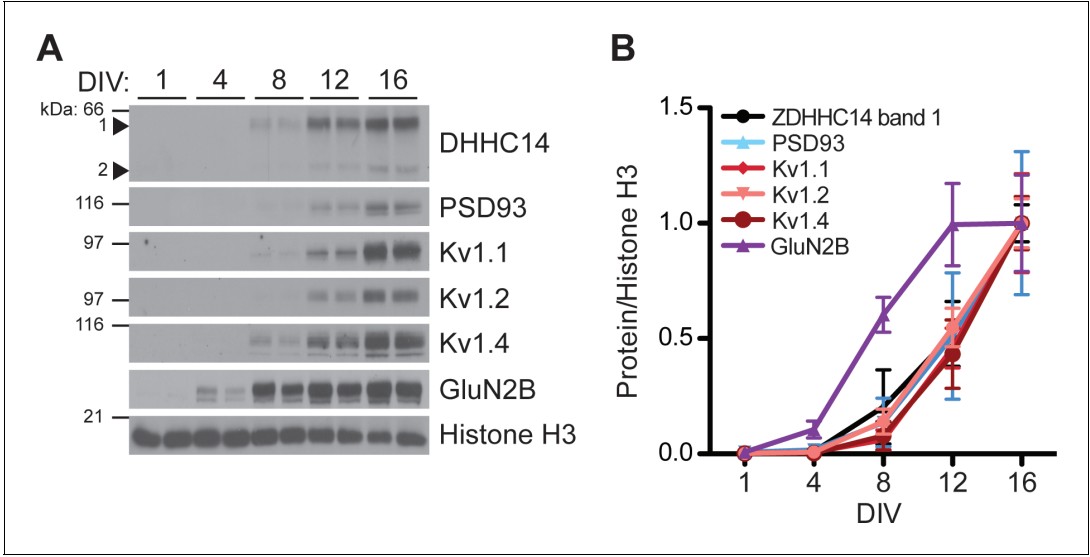

**Figure 5.** Developmental expression of ZDHHC14 mirrors that of PSD93 and Kv1-type potassium ion channels in cultured hippocampal neurons. (**A**) Hippocampal neurons were harvested at the indicated days in vitro (DIV) and lysates were western blotted with the indicated antibodies. (**B**) Quantified levels of the indicated proteins from *A*, relative to Histone H3 and normalized to DIV16 condition, plotted as a function of DIV (N = 3–4). Uncropped western blot images are in *Figure 5—figure supplement 2*.

The online version of this article includes the following source data and figure supplement(s) for figure 5:

**Source data 1.** Source data for *Figure 5B*.
**Figure supplement 1.** Kv1 channel palmitoylated cysteine residues and surrounding regions are conserved.
**Figure supplement 2.** Uncropped western blot images for *Figure 5*.

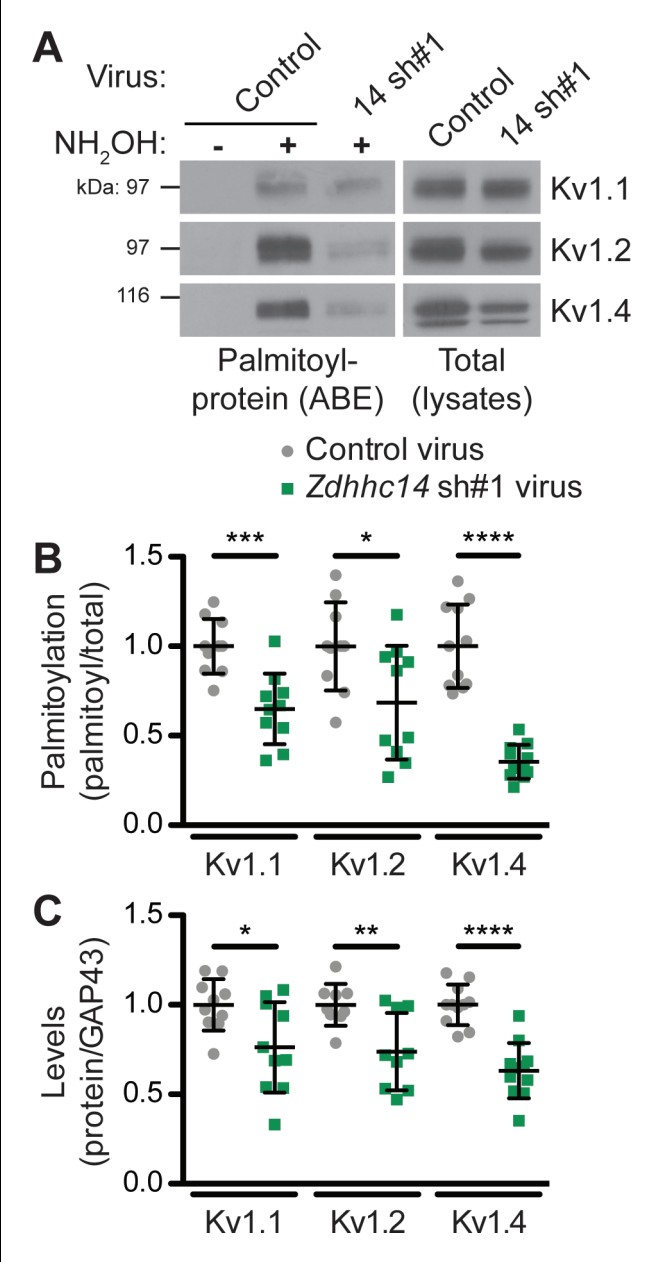

**Figure 6.** ZDHHC14 palmitoylates Kv1-type potassium channels in hippocampal neurons. (**A**) Hippocampal neurons were transduced with the indicated lentiviruses on DIV9 and harvested on DIV16 to assess Kv1 channel palmitoylation by ABE. Palmitoyl-proteins (isolated by ABE; *left panels*) and total protein levels (in parent lysates; *right panels*) were assessed by western blotting with the indicated antibodies. (**B**) Quantified palmitoyl:total levels of Kv1.1, Kv1.2, and Kv1.4 from *A*, normalized to the control virus condition (Kv1.1: unpaired Student's t-test ***p=0.0003, N = 10, 95% CI [0.18, 0.52]; Kv1.2: unpaired Student's t-test *p=0.024, N = 10, 95% CI [0.047, 0.58]; Kv1.4: Mann Whitney test ****p<0.0001, N = 11, U = 0.0). (**C**) Quantified total levels of Kv1.1 (*left*), Kv1.2 (*middle*), and Kv1.4 (*right*) from *A*, normalized to the control virus condition (Kv1.1: unpaired Student's t-test *p=0.019, N = 10, 95% CI [0.044, 0.43]; Kv1.2: unpaired Student's t-test **p=0.0058, N = 9, 95% CI [0.087, 0.43]; Kv1.4: unpaired Student's t-test ****p<0.0001, N = 11, 95% CI [0.25, 0.49]). Uncropped western blot images are in *Figure 6—figure supplement 1*.

The online version of this article includes the following source data and figure supplement(s) for figure 6:

**Source data 1.** Source data for *Figure 6B,C*.
**Figure supplement 1.** Uncropped western blot images for *Figure 6*.

## ZDHHC14 is required for targeting of Kv1 channels to the AIS

Given that ZDHHC14 is required for palmitoylation of Kv1 channels and their AIS scaffold, PSD93, we asked whether ZDHHC14 might also regulate AIS targeting of Kv1 channels themselves. All three Kv1 channels were readily detected at the AIS in neurons infected with control lentivirus (*Figure 7A, E,I*). However, in *Zdhhc14* knockdown neurons, AIS targeting of Kv1.1 (*Figure 7B,C,D*), Kv1.2 (*Figure 7F,G,H*), and Kv1.4 (*Figure 7J,K,L*) was reduced. We note that the total amount of AIS-localized Kv1.2 was significantly reduced, even though the reduction in Kv1.2 mean intensity at the AIS did not reach statistical significance (*Figure 7G,H*). Additional examples of reduced AIS localization of Kv1 channels after *Zdhhc14* knockdown are shown in *Figure 7—figure supplement 1*. These findings suggest that ZDHHC14 is required for AIS targeting of Kv1 channel subunits.

## ZDHHC14 is predominantly a Golgi-localized PAT in hippocampal neurons

We next asked whether ZDHHC14 more likely palmitoylates PSD93 and/or Kv1 channels directly at the AIS, or palmitoylates them earlier in their trafficking route, for example within the Golgi apparatus, where many PATs are localized (*Ernst et al., 2018*; *Ohno et al., 2006*). Interestingly, we found that HA-ZDHHC14 localized predominantly to the somatic Golgi and to punctate structures in dendrites (*Figure 8A–C*). ZDHHC14 was also detectable in axons but did not enrich at the AIS (*Figure 8C*). These findings increase the likelihood that ZDHHC14-dependent palmitoylation of PSD93 and Kv1 channels occurs within the Golgi, rather than directly at the AIS.

## Loss of ZDHHC14 reduces outward currents and increases action potential firing in hippocampal neurons

Finally, we addressed whether loss of ZDHHC14 alters neuronal physiological function, and the extent to which any such alterations might be consistent with changes in AIS-localized Kv1 channels. Indeed, we found that the density of outward currents, which are likely mediated by voltage-dependent potassium channels, was dramatically decreased in *Zdhhc14* knockdown neurons (*Figure 9A–C*). To determine whether these outward current changes might be linked to changes in AIS function we assessed action potential (AP) generation, and the properties of individual APs, in *Zdhhc14* knockdown neurons. Kv1-type channels are low-threshold, fast-activating and thus play an important role at the AIS in modulating AP firing by suppressing generation of, and shortening, APs (*Dodson et al., 2002*; *Goldberg et al., 2008*; *Yamada and Kuba, 2016*). Consistent with a reduced Kv1 number and/or function, *Zdhhc14* knockdown neurons fired a markedly greater number of APs in response to injection of a given current (*Figure 9D–F*). Moreover, the latency to the first action potential, rheobase (the minimum current required to generate an AP), and the rate of AP repolarization were all decreased in *Zdhhc14* knockdown neurons (*Figure 9D–E,G*, *Table 1*). The overall increased excitability and the individual changes in AP properties are all consistent with a loss of AIS-localized Kv1 channels in the absence of ZDHHC14. We also observed a slightly increased input resistance and a slightly depolarized resting membrane potential in *Zdhhc14* knockdown neurons (*Table 1*). Although also potentially accounted for by direct Kv1 loss, these latter changes could be downstream secondary consequences of Kv1 loss or could be due to alterations in other voltage-dependent channels in the absence of ZDHHC14.

## Discussion

Precise control of ion channel and receptor subcellular localization is critical to regulate and adjust neuronal excitability. Palmitoylation is a key mechanism that allows precise and dynamic regulation of protein targeting during development and in response to neuronal input (*Sanders et al., 2015*). However, while roles of palmitoylation in regulating protein targeting at the synapse are reasonably well described (*Brigidi et al., 2014*; *Firestein et al., 2000*; *Fukata and Fukata, 2010*; *Fukata et al., 2013*; *Hayashi et al., 2005*; *Hayashi et al., 2009*), far less is known regarding the importance of palmitoylation at other sites involved in neuronal excitation. Here we identify ZDHHC14 as a key controller of palmitoylation and AIS targeting of PSD93 and Kv1 family channels. Loss of ZDHHC14 also profoundly increases neuronal excitability, a finding that could be explained by loss of Kv1 channels from the AIS. Clustering of Kv1 channels at the AIS is thus controlled by palmitoylation-dependent

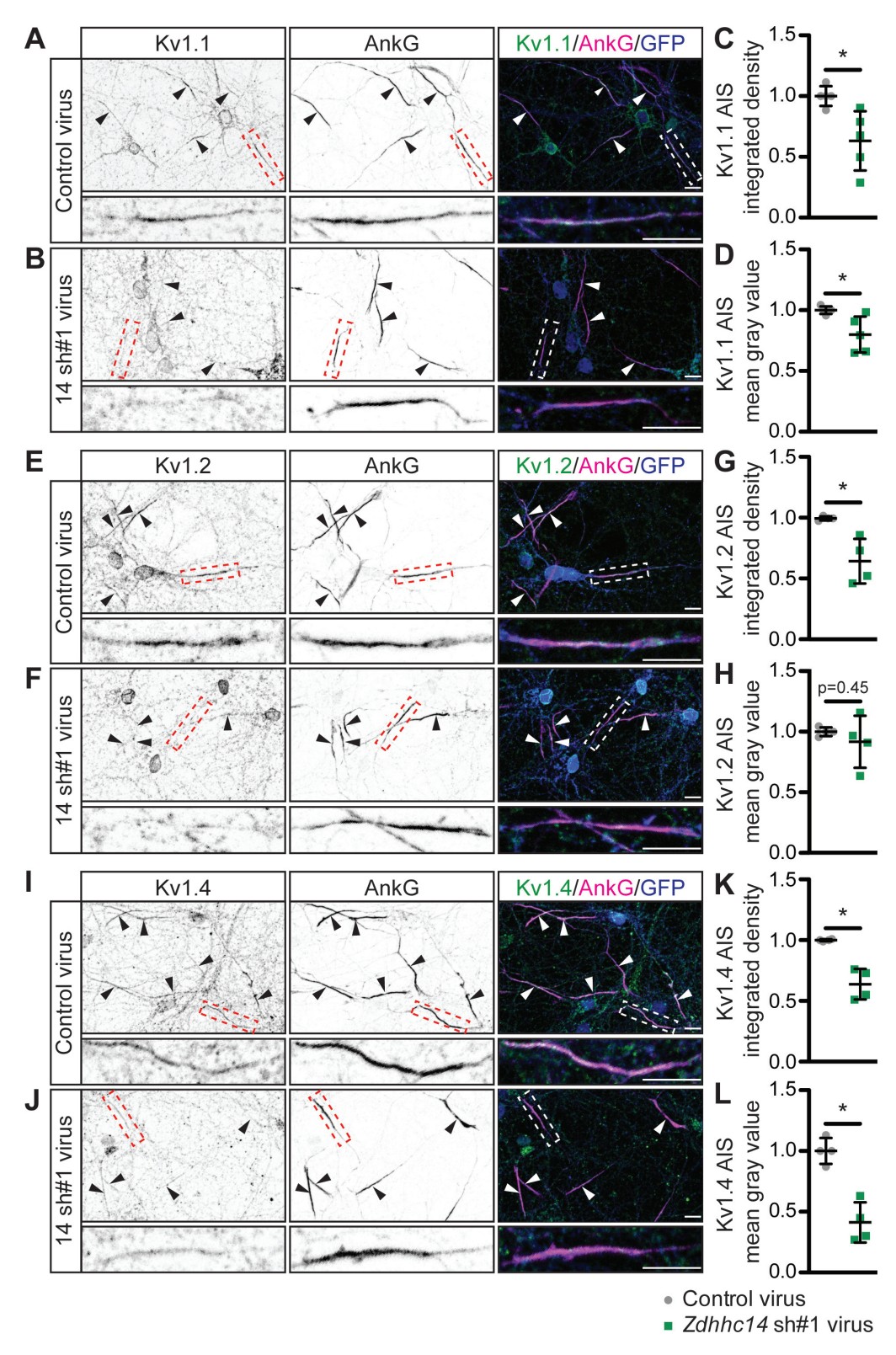

**Figure 7.** ZDHHC14 is required for Kv1-type potassium channel targeting to the AIS in hippocampal neurons. Hippocampal neurons were transduced on DIV9 with lentiviruses to express GFP without (Control; **A**) or with *Zdhhc14* shRNA#1 (**B**) and fixed on DIV16 to detect Kv1.1 (*left column* and green in merged images on right), AnkG (AIS marker, *middle column* and magenta in merged images on right) and GFP (marker of infected neurons, blue in merged images, *right column*). Lower panels show magnified view of red or white dashed boxed area in upper panels. (**C** and **D**) integrated density

*Figure 7 continued on next page*

*Figure 7 continued*

and mean gray value, respectively, of Kv1.1 signal within AnkG-defined AIS from images from *A*, normalized to control condition (*C*: Mann Whitney test *p=0.020, N = 3 independent cultures, U = 1.00). (*D*: Mann Whitney test *p=0.020, N = 3 independent cultures, U = 1.00). (**E** and **F**) as in *A* and *B*, except that antibodies were used to detect Kv1.2, AnkG, and GFP. (**G** and **H**) as in *C* and *D* but for Kv1.2 AIS targeting from images from *E* and *F* (*G*: Mann Whitney test p=0.029, N = 4 independent cultures, U = 0.00; *H*: Mann Whitney test p=0.45, N = 4 independent cultures, U = 5.00). (**I** and **J**) as in *A* and *B*, except that antibodies were used to detect Kv1.4, AnkG, and GFP. (**K** and **L**) as in *C* and *D* but for Kv1.4 AIS targeting from images from *I* and *J* (*K*: Mann Whitney test *p=0.029, N = 3 independent cultures, U = 0.0; *L*: Mann Whitney test *p=0.029, N = 3 independent cultures, U = 0.0). Scale bar in full and magnified views: 10 µm. Arrowheads in all panels indicate additional non-magnified AISs in the image. An additional set of example images is provided in *Figure 7—figure supplement 1*.

The online version of this article includes the following source data and figure supplement(s) for figure 7:

**Source data 1.** Source data for *Figure 7C,D,G,H,K and L*.

**Figure supplement 1.** Additional example images showing that ZDHHC14 is required for Kv1-type potassium channel targeting to the AIS in hippocampal neurons.

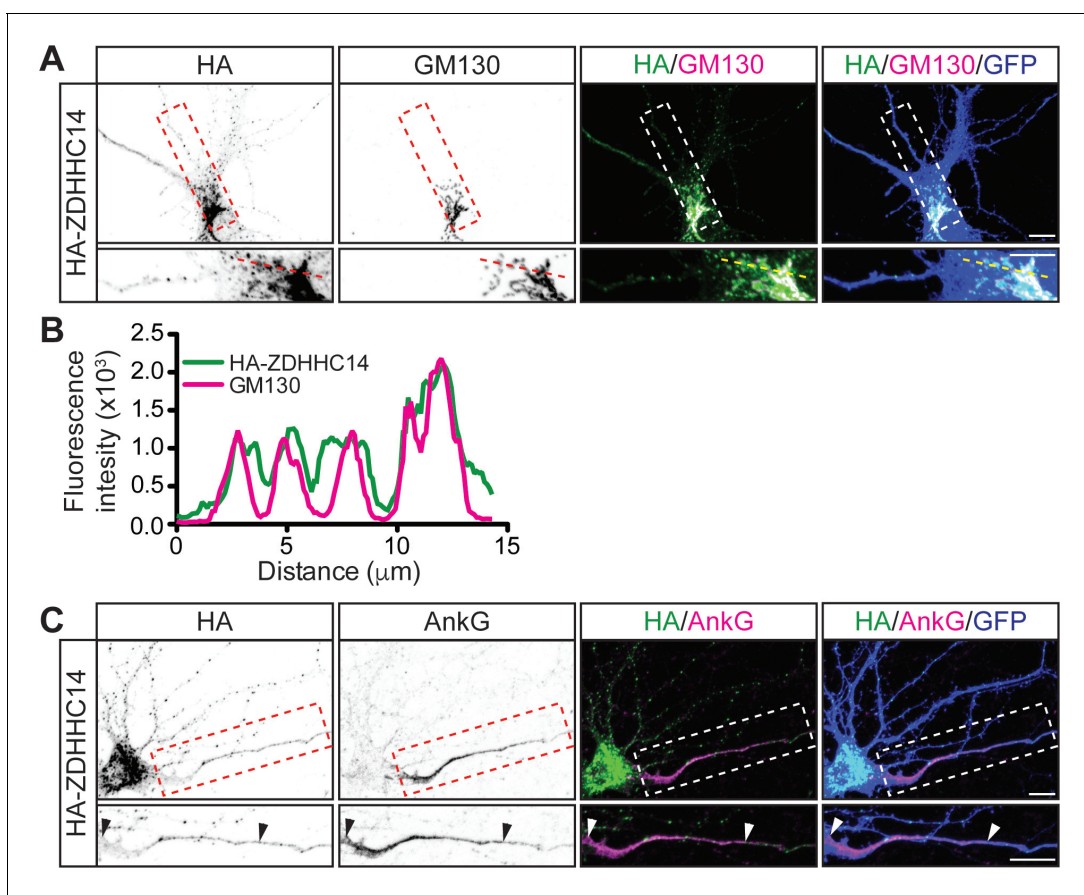

**Figure 8.** ZDHHC14 predominantly localizes to the Golgi. (**A**) Hippocampal neurons were transfected to express GFP and HA-ZDHHC14. Neurons were then fixed and immunostained with antibodies against HA (*left column* and green in merged images), GM130 (Golgi marker, *second column* and magenta in merged images), and GFP (marker of infected neurons, blue in merged, *right column*). Magnified views of red or white dashed boxed area of top row images are shown below. (**B**) Graph of the fluorescent intensity profiles of HA-ZDHHC14 (green) and GM130 (red) along the red or yellow dotted line indicated in the lower panels of *A*. (**C**) Neurons transfected as in *A* were immunostained with antibodies against HA (*left column* and green in merged images), AnkG (AIS marker, *second column* and magenta in merged images), and GFP (marker of infected neurons, blue in merged, *right column*). Magnified views of boxed area of top row images are shown below and arrowheads indicate the start and end of the AIS. Scale bar: 10 µm.

The online version of this article includes the following source data for figure 8:

**Source data 1.** Source data for *Figure 8B*.

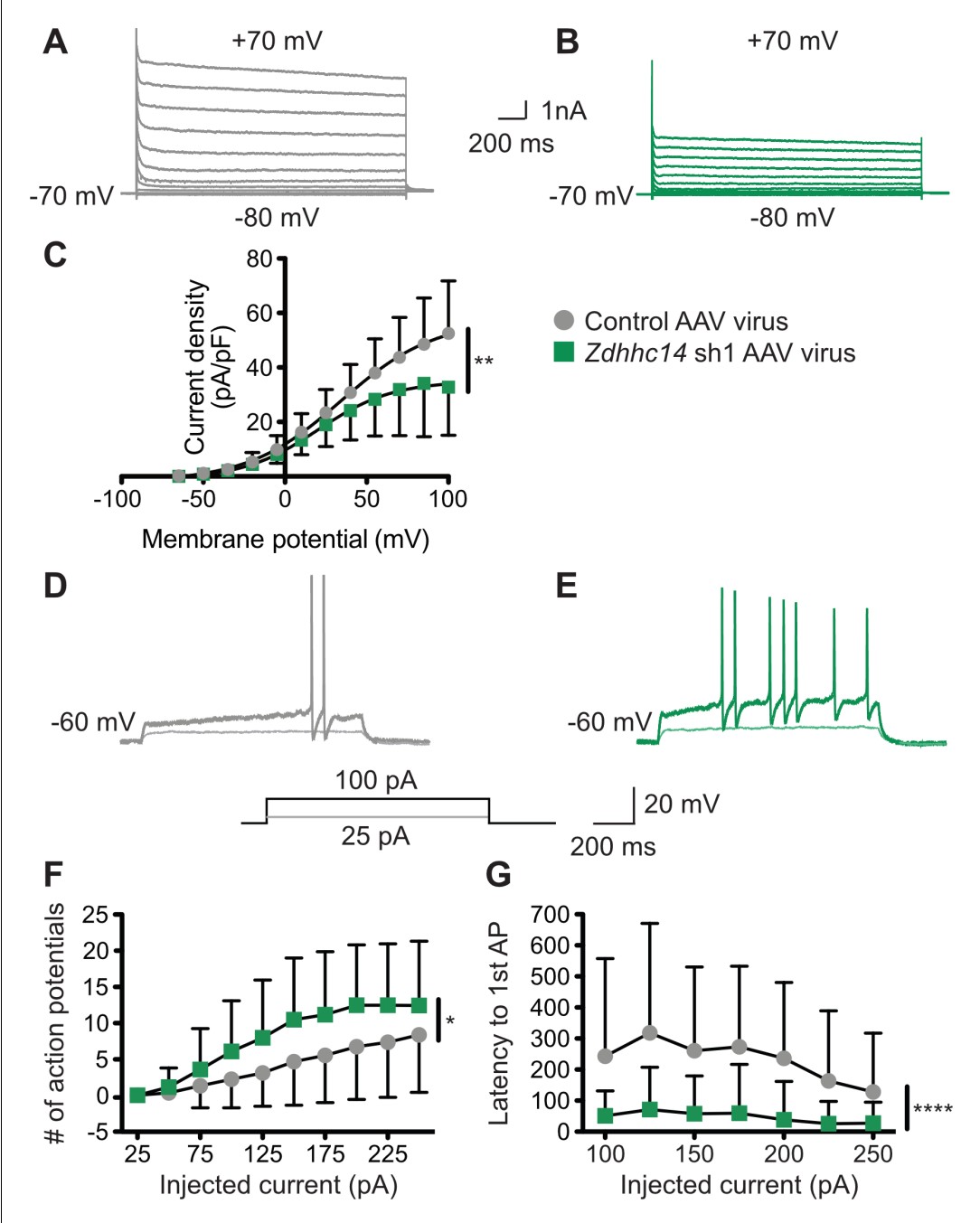

**Figure 9.** Loss of ZDHHC14 increases excitability of hippocampal neurons. (A, B) Hippocampal neurons were transduced with AAV to express GFP alone (A) or GFP plus *Zdhhc14* shRNA#1 (B) and GFP-positive cells were subjected to whole-cell patch-clamp. Representative traces showing the development of outward currents following voltage steps from −80 to +100 mV (Δ15 mV). (C) Summary graph of outward current density following steps of the indicated voltages for neurons infected with control AAV (gray circles) or *Zdhhc14* sh#1 AAV (green squares). Zdhhc14 knockdown reduces outward current density (C) Repeated Measures ANOVA, Virus p=0.18 [F(1)=2.06], membrane potential p<0.0001 [F(12)=73.14], **interaction p=0.0027 [F(12)=2.68]; (N = 7). There was also a significant difference (p<0.0001 [F(4)=8.46]) between conditions when the two curves were fitted with a Boltzmann equation. (D, E) Membrane potential was kept at −60 to −65 mV mV by injecting a small DC current through the recording pipette and voltage responses to the indicated current injection steps were measured. Representative traces from control (D) and *Zdhhc14* shRNA#1 (E) transduced neurons in response to the indicated current injection steps are shown. (F, G) Summary graphs of number of action potentials fired (F) and latency to first action potential (AP, G) following injection of the indicated currents for neurons infected with control AAV (gray circles) or *Zdhhc14* sh#1 AAV (green squares). *Zdhhc14* knockdown increases AP firing (F: Repeated Measures ANOVA, *Virus p=0.020 [F(1)=5.84], injected current p<0.0001 [F(9)=49.31], interaction

*Figure 9 continued on next page*

*Figure 9 continued*

p=0.0002 [F(9)=3.64]; N = 23) and decreases latency to first AP (*G*: mixed effects model analysis, \*\*\*\*Virus p<0.0001 [F(1,43)=19.99], injected current p<0.0001 [F(3.35, 116.70)=15.18], interaction p<0.0001 [F(6, 209)=6.30]; N = 23).

The online version of this article includes the following source data for figure 9:

**Source data 1.** Source data for *Figure 9C,F and G*.

mechanisms reminiscent of those that govern targeting glutamate receptors to synapses (*Firestein et al., 2000*; *Fukata et al., 2013*; *Hayashi et al., 2005*; *Hayashi et al., 2009*; *Thomas et al., 2012*; *Thomas and Huganir, 2013b*). However, AIS targeting of Kv1 channels involves a distinct PDZ-domain protein and a different, largely unstudied PAT.

PSD93's interaction with, and palmitoylation by, ZDHHC14 is dependent on ZDHHC14's C-terminal PDZ ligand (*Figure 1*), suggesting that ZDHHC14 recognizes PSD93, and perhaps other substrates, in a PDZ ligand-dependent manner. This PDZ ligand-dependent mechanism of substrate recognition by a PAT was also reported for ZDHHC5 and ZDHHC8, which use their Type-II PDZ ligands to bind and palmitoylate the PDZ domain-containing protein GRIP1b (*Thomas et al., 2012*). While these are the only three PATs known to use this mechanism of substrate recognition, certain kinases also bind and phosphorylate PDZ domain-containing substrates in a manner dependent on the kinase's PDZ ligand (*Hasegawa et al., 1999*; *Sabio et al., 2004*; *Thomas et al., 2005*).

Although the C-termini of Kv1.1, Kv1.2, and Kv1.4 are not predicted to bind ZDHHC14, they all terminate in Type-I PDZ ligands that (i) are critically important for their axonal targeting (*Arnold and Clapham, 1999*) and (ii) can directly bind PSD93 (*Kim et al., 1995*; *Kim et al., 1996*). Thus, it is possible that PSD93 is not only a direct ZDHHC14 substrate but also acts as a scaffold to allow ZDHHC14 to control palmitoylation and subcellular targeting of Kv1 channels. This possibility is further increased by our finding that ZDHHC14 binds PDZ domain 3 of PSD93 (*Figure 1—figure supplement 2*), while Kv1.4 binds PDZ domain 2 of PSD93 (originally described as an unidentified 'Clone 5' by *Kim et al., 1995*) and subsequently confirmed as PSD93/Chapsyn-110 by *Kim et al., 1996*. However, ZDHHC14 is not enriched at the AIS, but rather predominantly localizes to the Golgi (*Figure 8A,B*). These findings would support a model in which PSD93 acts as a scaffold for Kv1

**Table 1.** Electrophysiological properties of control and Zdhhc14 sh#1 (14 sh#1) AAV- infected hippocampal neurons.

| | Control AAV | 14sh#1 AAV | p-value |
|---|---|---|---|
| $V_{hold}$ (mV) | −63.14 +/− 1.53 | −63.13 +/− 1.30 | 0.67[MW] |
| Rheobase (mA) | 140.22 +/− 75.84 | 86.96 +/− 39.07 | \*\*0.0052[MW] |
| Peak amplitude (mV) | 103.33 +/− 7.66 | 87.41 +/− 10.00 | \*\*\*\*0.00000028 |
| Half width (ms) | 2.68 +/− 0.50 | 3.02 +/− 0.94 | 0.18[MW] |
| Max rise slope (mV/ms) | 154.15 +/− 49.03 | 89.76 +/− 44.78 | \*\*\*\*<0.0001[MW] |
| Max decay scope (mV/ms) | −42.21 +/− 8.39 | −33.16 +/−9.90 | \*\*0.0017 |
| Input resistance: −25 pA | 324.86 +/− 144.03 | 393.27 +/− 189.20 | 0.17 |
| −50 pA | 326.76 +/− 146.95 | 409.55 +/− 160.52 | 0.075 |
| −75 pA | 319.91 +/− 132.03 | 411.04 +/− 163.64 | \*0.044 |
| −100 pA | 304.48 +/− 113.12 | 397.83 +/− 160.32 | \*0.027 |
| Peak dV/dt | 133.62 +/− 40.47 | 85.45 +/− 41.07 | \*\*\*0.0002 |
| Anti-peak dV/dt | −37.62 +/− 7.99 | −32.11 +/− 9.56 | \*0.038 |
| Resting membrane potential (mV) | −60.51 +/− 8.08 | −54.79 +/− 8.31 | \*0.022 |

Mean +/− SD is shown; N = 23; data were analyzed using unpaired Student's t-test unless otherwise indicated: [MW]=Mann Whitney test.

The online version of this article includes the following source data for Table 1:

**Source data 1.** Source data for *Table 1*.

channels and ZDHHC14 at the Golgi to allow ZDHHC14-dependent palmitoylation of Kv1 channels and their subsequent co-trafficking with PSD93 to the AIS. This mechanism is not unprecedented, as palmitoylation within the Golgi was reported to facilitate forward trafficking of the BK potassium channel (*Tian et al., 2012*).

ZDHHC14 loss significantly reduces palmitoylation of PSD93 and Kv1 channels (*Figure 2*, *Figure 6*) but does not eliminate palmitoylation of these proteins. Whether this residual palmitoylation is due to the presence of other PATs, incomplete *Zdhhc14* knockdown, or both, is currently unknown. Assessing residual (non-ZDHHC14-dependent) palmitoylation of these proteins could potentially require developing reagents to knock-down multiple additional PATs. Some potential insight into this issue is provided by a recent report that the zebrafish ortholog of Kv1.1 could be palmitoylated by the PAT HIP14 (ZDHHC17) in cotransfected HEK293 cells, although palmitoylation of endogenous neuronal Kv1.1 by ZDHHC17/HIP14 was not assessed (*Nelson et al., 2020*). HIP14/ZDHHC17 might thus be a candidate to contribute to residual palmitoylation of Kv1.1 (as well as possibly other Kv1 channels). However, ZDHHC17 is less likely to mediate residual PSD93 palmitoylation because this PAT lacks a Type-I PDZ ligand and loss of ZDHHC17 does not affect PSD93 palmitoylation in mice (*Wan et al., 2013*). While we cannot rule out roles for ZDHHC17/HIP14 and/or other PATs, our findings suggest that PSD93 and Kv1 palmitoylation is dominantly controlled by ZDHHC14 in hippocampal neurons.

The temporally similar developmental upregulation of ZDHHC14, PSD93, and Kv1 channel protein levels (*Figure 5*) is striking, but such a biochemical analysis cannot address the subcellular location(s) to which these proteins localize as their levels increase. However, a prior study reported, albeit non-quantitatively, that Kv1.2 is first detectably enriched at the AIS of cultured hippocampal neurons at a very similar timepoint (DIV13) to when we observe marked upregulation of total Kv1.2 protein levels (DIV12; *Figure 5*; *Sánchez-Ponce et al., 2012*). This finding strengthens the possibility that Kv1.2 upregulation coincides with and/or results in its AIS targeting. We nonetheless appreciate that more in-depth analyses, potentially involving laser-capture dissection of sufficient individual AIS's to run immunoblots and/or comprehensive, semi-quantitative immunocytochemical experiments at individual AIS's, would be required to further test this hypothesis.

Intriguing, additional support for a shared function of ZDHHC14, PSD93, and Kv1 channels in AIS regulation emerges from examining the evolutionary conservation of ZDHHC14, PSD93, and the AIS itself. Sodium channel clustering in proximal axons is first observed immunohistologically in vertebrates, coincident with the evolution of AIS-anchoring domains in these channels (*Kole and Stuart, 2012*). Similarly, N-terminal palmitoylation sites are only present in vertebrate orthologs of PSD93 and are absent in the fruit fly PSD93 ortholog Discs Large (DLG) (*Thomas and Hayashi, 2013a*). Lastly, ZDHHC14's PDZ ligand, which greatly increases ZDHHC14's ability to palmitoylate PSD93 (*Figure 1*), is first observed in simple chordates (*Ciona intestinalis*) and vertebrates (*Figure 1—figure supplement 1*). In contrast, palmitoyl-sites and PDZ ligands in Kv1 channels are more evolutionarily ancient (*Figure 5—figure supplement 1B*). However, the ability of ZDHHC14 to palmitoylate PSD93, and for palmitoyl-PSD93 to act as a scaffold to cluster Kv1 channels at the AIS, appear to have evolved with the appearance of the AIS itself as nervous systems became more complex.

Several lines of evidence are consistent with direct loss of Kv1-type voltage-gated potassium channels accounting for the increased excitability of *Zdhhc14* knockdown neurons. First, ZDHHC14 loss decreases outward currents, a phenotype most readily accounted for by reduced number and/or function of voltage-gated potassium channels (*Figure 9*; *Dodson et al., 2002*). ZDHHC14 loss also reduces AIS localization of multiple Kv1 family channels, assessed immunocytochemically (*Figure 7*). Moreover, Kv1-type channels activate at subthreshold membrane potentials and counteract voltage-gated sodium channel (Nav) currents, thereby dampening action potential firing and reducing neuronal excitability (*Clark et al., 2009*; *Dodson et al., 2002*; *Yamada and Kuba, 2016*). The decrease in outward currents and increase in excitability seen after ZDHHC14 loss are thus both consistent with a reduction in Kv1 channel number and/or function.

Specific features of APs in *Zdhhc14* knockdown neurons are also consistent with reduced Kv1 channel number and/or function. For example, Kv1-type channels are known to regulate AP firing by suppressing generation of, as well as shortening, action potentials (*Yamada and Kuba, 2016*) so Kv1 channel loss at the AIS could account for the decreases in latency to the first action potential, rheobase, and rate of action potential repolarization seen in *Zdhhc14* knockdown neurons (*Figure 9D,E, G*; *Table 1*). Indeed, the slight but significant AP broadening that we observed in *Zdhhc14*

knockdown neurons (half-width; *Table 1*) was also seen following acute block of Kv1 channels (*Guan et al., 2018*; *Kole et al., 2007*).

We are nonetheless aware that other factors may also influence excitability in *Zdhhc14* knockdown neurons. For example, while the slightly increased resting membrane potential is also consistent with loss of Kv1 channels, we cannot exclude roles for other voltage-gated potassium channels in this increase. An increased resting membrane potential could also inactivate a subset of voltage-gated sodium channels, which might in turn account for the decreased peak AP amplitude and rate of AP rise seen with *Zdhhc14* knockdown (*Table 1*).

Another change seen after ZDHHC14 loss is the shortening of the AIS itself (*Figure 3D*), previously reported to be a compensatory response to increased neuronal excitability (*Dumitrescu et al., 2016*; *Grubb et al., 2011*; *Grubb and Burrone, 2010*). Like the increased membrane potential, this decreased AIS length could thus be a secondary consequence of Kv1 loss after *Zdhhc14* knockdown but could also be due to other changes. In summary, several aspects of the excitability changes in *Zdhhc14* knockdown neurons are consistent with a direct loss of Kv1 channels, but we recognize that other changes may be downstream secondary consequence of such loss, or may be due to altered regulation of other voltage-dependent channels and/or AIS structural proteins.

We also recognize that mutating the palmitoyl-cysteines of the PSD93β isoform reduces AIS localization of PSD93 (*Figure 4*) to a lesser extent than is seen following loss of ZDHHC14 itself (*Figure 3*). These findings suggest that additional ZDHHC14 substrate(s) may also contribute to AIS localization of PSD93. Kv1 channels themselves, which PSD93 directly binds and which are robustly palmitoylated in a ZDHHC14-dependent manner (*Figure 6*), could potentially play such a role. It is especially intriguing that many other AIS-localized ion channels, scaffolds, and cell adhesion molecules are either known or predicted to be palmitoylated (*Blanc et al., 2015*; *He et al., 2012*; *Nelson and Jenkins, 2017*; *Pei et al., 2016*). It would be important to determine in the future the extent to which ZDHHC14 loss impairs palmitoylation of these proteins, and how any such additional changes in palmitoylation might affect not just PSD93 localization, but also the broader structure and/or function of the AIS.

Given that ZDHHC14 may regulate other palmitoyl-proteins, it is theoretically possible that signals in our biochemical and/or immunocytochemical experiments could represent heteromultimers of PSD93 and its paralogous palmitoyl-protein PSD95, which were reported in transfected non-neuronal cells (*Kim et al., 1996*). However, subsequent studies of endogenous proteins from brain tissue suggested that although both PSD95 and PSD93 are found in multimeric complexes, endogenous PSD93/PSD95 heteromultimers were not detected (i.e. the endogenous multimeric complexes consist of homomeric PSD95 or PSD93, but not both) (*Hsueh et al., 1997*). Moreover, any potential heteromultimerization would not be an issue for our ABE assays, which assess denatured proteins that run as monomers on SDS-PAGE (*Figure 1*, *Figure 2*). We also note that our PSD93 antibody is validated using knockout tissue by the supplier. This antibody might still theoretically detect PSD95 in immunocytochemical experiments, but a prior study reported that PSD95 (detected with another widely used monoclonal antibody, which is itself validated using knockout tissue) is undetectable at the AIS, whereas PSD93 is AIS-localized (*Ogawa et al., 2008*). Importantly, we used the same PSD93 antibody, and a very similar fixation protocol, to the work of Ogawa et al. We would thus expect that antibodies against PSD93 (and also PSD95, if we were to use them) would show similar specificity in our experiments. Together, these findings suggest that we do not detect PSD93/PSD95 heteromultimers in either biochemical or immunocytochemical assays.

We also note that the effect of ZDHHC14 loss on neuronal excitability is more dramatic than the effect on AIS targeting of any individual Kv1 channel (*Figure 7* versus 9). One possible explanation for this finding is that palmitoylation not only regulates channel targeting to the AIS but also regulates Kv1 channel electrophysiological responses. Indeed, palmitoylation was previously shown to modulate voltage-sensing of Kv1.1 expressed in Sf9 cells and loss of Kv1.1 palmitoylation decreased current amplitude (*Gubitosi-Klug et al., 2005*). Importantly, though, another intriguing possibility is that ZDHHC14 is a master regulator of multiple AIS ion channels and/or scaffold proteins and that impaired localization and/or function of these additional ZDHHC14 substrates further impacts neuronal excitability.

While we have focused this study on physiological neuronal regulation, our findings also have implications for our understanding of how neuronal excitability is altered in conditions such as epilepsy. Indeed, mutations in multiple Kv1 channel genes and copy number variations of the *Dlg2*

gene (which codes for PSD93) are associated with multiple neurological conditions characterized by neuronal hyperexcitability, including epilepsy, myokemia, episodic ataxia, schizophrenia, autism spectrum disorder, and intellectual disability (*Cuscó et al., 2009*; *Eunson et al., 2000*; *Felix, 2000*; *Gao et al., 2018*; *Reggiani et al., 2017*; *Walsh et al., 2008*). It is an intriguing possibility that dysregulated palmitoylation of PSD93 and/or Kv1 channels contributes to the pathogenesis of such conditions. In contrast, ZDHHC14 has not, thus far, been associated with any neuropathological condition. However, ZDHHC14 is one of only four PATs intolerant to loss of function mutations in humans, suggesting that it may be an essential gene (*Lek et al., 2016*). While it is possible that roles of ZDHHC14 in non-neuronal tissues underlie this requirement, it is again intriguing that ZDHHC14's potential action as a master regulator of AIS structure and function could account for its indispensability.

In summary, our findings that palmitoylation of PSD93 by ZDHHC14 is important for regulating targeting of voltage-gated ion channels to the AIS and for neuronal excitability, provide key insights into multiple outstanding questions. To our knowledge, these findings identify the first substrates of ZDHHC14 in any system and reveal, for the first time, a role for PSD93 palmitoylation in neurons, a question that has remained unanswered for two decades. We also demonstrate that Kv1 family voltage-gated potassium channels are endogenously palmitoylated in neurons and provide additional mechanistic insight into the regulation of their AIS targeting and physiology. These findings have important implications for our understanding of physiological regulation of neuronal excitability and its dysfunction in epilepsy and/or other conditions marked by hyper-excitation.

# Materials and methods

## Key resources table

| Reagent type (species) or resource | Designation | Source or reference | Identifiers | Additional information |
|---|---|---|---|---|
| Other | ZDHHC14 | GenBank | NP_078906.2 | Protein (*Homo sapiens*) |
| Other | ZDHHC14 | GenBank | NP_666185.3 | Protein (*Mus musculus*) |
| Other | ZDHHC14 | GenBank | NP_001034432.1 | Protein (*Rattus norvegicus*) |
| Other | ZDHHC14 | GenBank | XP_004914695.1 | Protein (*Xenopus tropicalis*) |
| Other | ZDHHC14 | GenBank | XP_005160409.1 | Protein (*Danio rerio*) |
| Other | ZDHHC14 | GenBank | XP_002127630.1 | Protein (*Ciona intestinalis*) |
| Other | app (*approximated*) | GenBank | NP_001137937.1 | Protein (*Drosophila melanogaster*) |
| Other | DHHC-2 | GenBank | NP_0493007.2 | Protein (*Caenorhabditis elegans*) |
| Other | PSD93β (*Dlg2* gene product) | GenBank | XP_017445141.1 | Protein (*R. norvegicus*) |
| Other | PSD93β | GenBank | NP_001338205.1 | Protein (*H. sapiens*) |
| Other | Kv1.1 | GenBank | NP_775118.1 | Protein (*R. norvegicus*) |
| Other | Kv1.2 | GenBank | NP_037102.1 | Protein (*R. norvegicus*) |
| Other | Kv1.4 | GenBank | NP_037103.1 | Protein (*R. norvegicus*) |

*Continued on next page*

*Continued*

| Reagent type (species) or resource | Designation | Source or reference | Identifiers | Additional information |
|---|---|---|---|---|
| Other | Kv1.1 | GenBank | NP_000208.2 | Protein (*H. sapiens*) |
| Other | Kv1.1 | GenBank | NP_034725.3 | Protein (*M. musculus*) |
| Other | Kv1.1 | GenBank | XP_004912858.1 | Protein (*X. tropicalis*) |
| Other | Kv1.1 | GenBank | XP_005163101.1 | Protein (*D. rerio*) |
| Other | Sh (*Shaker*) | GenBank | NP_523393.3 | Protein (*D. melanogaster*) |
| Other | SHK-1 | GenBank | NP_871935.1 | Protein (*C. elegans*) |
| Recombinant DNA reagent | FEW-*PSD93α*-myc | This paper | Source: Thomas Lab | Lentiviral construct to transfect (HEK293 cells and rat neurons) and express *R. norvegicus* cDNA |
| Recombinant DNA reagent | FEW-*PSD93β*-myc | This paper | Source: Thomas Lab | Lentiviral construct to transfect (HEK293 cells and rat neurons) and express *R. norvegicus* cDNA |
| Recombinant DNA reagent | FEW-*PSD93β*-myc C10S | This paper | Source: Thomas Lab | Lentiviral construct to transfect (HEK293 cells and rat neurons) and express *R. norvegicus* cDNA |
| Recombinant DNA reagent | FEW-*PSD93β*-myc C16,18S | This paper | Source: Thomas Lab | Lentiviral construct to transfect (HEK293 cells and rat neurons) and express *R. norvegicus* cDNA |
| Recombinant DNA reagent | FEW-*PSD93β*-myc C10,16,18S | This paper | Source: Thomas Lab | Lentiviral construct to transfect (HEK293 cells and rat neurons) and express *R. norvegicus* cDNA |
| Recombinant DNA reagent | FEW-*PSD93β*-myc 5CS | This paper | Source: Thomas Lab | Lentiviral construct to transfect (HEK293 cells and rat neurons) and express *R. norvegicus* cDNA |
| Recombinant DNA reagent | FEW-HA-*Zdhhc14* | This paper | Source: Thomas Lab | Lentiviral construct to transfect (HEK293 cells and rat neurons) and express *M. musculus* cDNA |
| Recombinant DNA reagent | FEW-HA-*Zdhhc14* LSSE | This paper | Source: Thomas Lab | Lentiviral construct to transfect (HEK293 cells and rat neurons) and express *M. musculus* cDNA |
| Recombinant DNA reagent | FUGW | Addgene | Cat #14883 (RRID:Addgene_14883) | Lentiviral construct to transduce rat neurons and express Human UbC-driven EGFP |

*Continued on next page*

*Continued*

| Reagent type (species) or resource | Designation | Source or reference | Identifiers | Additional information |
|---|---|---|---|---|
| Recombinant DNA reagent | FUGW H1-*Zdhhc14*sh#1 | This paper | Source: Thomas Lab | Lentiviral construct to transduce rat neurons and express Human UbC-driven EGFP and H1-driven shRNA (sequence: GCATTCAGA GCACCAAATTCGT) |
| Recombinant DNA reagent | FUGW H1-*Zdhhc14*sh#2 | This paper | Source: Thomas Lab | Lentiviral construct to transduce rat neurons and express Human UbC-driven EGFP and H1-driven shRNA (sequence: GCCACACTC TCAGACATTAT) |
| Recombinant DNA reagent | pAAV-GFP-*Zdhhc14*sh#1 | This paper | Source: Thomas Lab | AAV construct to transduce rat neurons and express EGFP and shRNA Backbone Addgene plasmid #26937 |
| Recombinant DNA reagent | pPC97 | *Dong et al., 1997* | | |
| Recombinant DNA reagent | pPC97 wild type Zdhhc14 C-term tail | This paper | Source: Thomas Lab | *Zdhhc14 M. musculus* gene fragment |
| Recombinant DNA reagent | pPC97 Zdhhc14 C-term tail LSSE | This paper | Source: Thomas Lab | *Zdhhc14 M. musculus* gene fragment |
| Recombinant DNA reagent | pPC86 | *Dong et al., 1997* | | |
| Recombinant DNA reagent | pPC86 PSD93 PDZ3 | This paper | Source: Thomas Lab | *Dlg2 (Psd93) M. musculus* gene fragment |
| Recombinant DNA reagent | pCIS GST-*Zdhhc14* C-term tail | This paper | Source: Thomas Lab | *Zdhhc14 M. musculus* gene fragment transfected in HEK293T cells |
| Recombinant DNA reagent | pCIS GST-*Zdhhc14* C-term tail LSSV | This paper | Source: Thomas Lab | *Zdhhc14 M. musculus* gene fragment transfected in HEK293T cells |
| Recombinant DNA reagent | pMDLg | Addgene | Cat #12251 (RRID:Addgene_12251) | Lentiviral Gag and Pol expressing plasmid |
| Recombinant DNA reagent | pRSV-Rev | Addgene | Cat #12253 (RRID:Addgene_12253) | Lentiviral Rev expressing plasmid |
| Recombinant DNA reagent | pMD2.G | Addgene | Cat #12259 (RRID:Addgene_12259) | Lentiviral VSV-G envelope expressing plasmid |
| Recombinant DNA reagent | pHelper | Agilent | Cat #240071 | |
| Recombinant DNA reagent | pAAV-RC | Agilent | Cat #240071 | |
| Strain, strain background (*S. cerevisiae*) | PJ69 | *James et al., 1996* | | |
| Strain, strain background (*S. cerevisiae*) | HF7C | *Feilotter et al., 1994* | | |

*Continued on next page*

*Continued*

| Reagent type (species) or resource | Designation | Source or reference | Identifiers | Additional information |
|---|---|---|---|---|
| Biological sample (*R. norvegicus*) | Primary hippocampal neurons | Charles River | | Freshly isolated from embryonic day 18 hippocampi |
| Antibody | Anti-PSD93 (mouse monoclonal IgG1) | NeuroMab | Cat #75–057 (RRID:AB_2277296) | WB (1:500) IF (1:100) |
| Antibody | Anti-Kv1.1 (mouse IgG2b) | NeuroMab | Cat #75–105 (RRID:AB_2128566) | WB (1:500) IF (1:100) |
| Antibody | Anti-Kv1.2 (mouse monoclonal IgG2b) | NeuroMab | Cat #75–008 (RRID:AB_2296313) | WB (1:500) IF (1:100) |
| Antibody | Anti-Kv1.4 (mouse monoclonal IgG1) | NeuroMab | Cat #75–010 (RRID:AB_2249726) | WB (1:500) IF (1:100) |
| Antibody | Anti-GluN2B (mouse monoclonal IgG2b) | NeuroMab | Cat #75–101 (RRID:AB_2232584) | WB (1:200) |
| Antibody | Anti-AnkG (mouse monoclonal IgG2a) | NeuroMab | Cat #75–146 (RRID:AB_10673030) | IF (1:200) |
| Antibody | Anti-GST (rabbit polyclonal) | Bethyl Laboratories | Cat #A190-122A (RRID:AB_67419) | WB (1:5000) |
| Antibody | Anti-HA tag (rabbit monoclonal) | Cell Signaling Technologies | Cat #3724 (RRID:AB_1549585) | WB (1:5000) |
| Antibody | Anti-Erk1/2 (rabbit monoclonal) | Cell Signaling Technologies | Cat #4696 (RRID:AB_390780) | WB (1:1000) |
| Antibody | Anti-Histone H3 (rabbit monoclonal) | Cell Signaling Technologies | Cat #4499 (RRID:AB_10544537) | WB (1:1000) |
| Antibody | Anti-Myc tag (rabbit monoclonal) | Cell Signaling Technologies | Cat #2278 (RRID:AB_490778) | WB (1:5000) IF (1:500) |
| Antibody | Anti-GAP43 (rabbit polyclonal) | Novus Biologicals | Cat #NB300-143 (RRID:AB_10001196) | WB (1:5000) |
| Antibody | Anti-GFP (chicken polyclonal) | Millipore-Sigma | Cat #AB16901 (RRID:AB_90890) | IF (1:500) |
| Antibody | Anti-GFP (rabbit polyclonal) | Thermo-Fisher Scientific | Cat #A-11122 | IF (1:1000) |
| Antibody | Sheep anti-mouse HRP-linked polyclonal | Millipore-Sigma | Cat #NA931 | WB (1:5000) |
| Antibody | Donkey anti-rabbit HRP-linked polyclonal | Jackson Immunoresearch | Cat #711–0350152 | WB (1:5000) |
| Antibody | AlexaFluor 488 goat anti-chicken polyclonal | Thermo Fisher Scientific | Cat #A-11039 (RRID:AB_142924) | IF (1:500) |
| Antibody | AlexaFluor 488 goat anti-rabbit polyclonal | Thermo Fisher Scientific | Cat #A-11032 (RRID:AB_2534091) | IF (1:500) |
| Antibody | AlexaFluor 568 goat anti-rabbit polyclonal | Thermo Fisher Scientific | Cat #A-11011 (RRID:AB_143157) | IF (1:500) |

*Continued on next page*

*Continued*

| Reagent type (species) or resource | Designation | Source or reference | Identifiers | Additional information |
|---|---|---|---|---|
| Antibody | AlexaFluor 568 goat anti-IgG2a polyclonal | Thermo Fisher Scientific | Cat #A-21134 (RRID:AB_2535773) | IF (1:500) |
| Antibody | AlexaFluor 647 goat anti-IgG2a polyclonal | Thermo Fisher Scientific | Cat #A-21241 (RRID:AB_141698) | IF (1:500) |
| Antibody | AlexaFluor 647 goat anti-IgG1 polyclonal | Thermo Fisher Scientific | Cat #A-21240 (RRID:AB_141658) | IF (1:500) |
| Antibody | AlexaFluor 647 goat anti-IgG2b polyclonal | Thermo Fisher Scientific | Cat #A-21242 (RRID:AB_2535811) | IF (1:500) |
| Antibody | Anti-ZDHHC14 rabbit polyclonal | This paper | Source: Thomas Lab | Immunogen: CDSLHEDSV RGLVKLSSV WB (1:200) |
| Chemical compound, drug | MMTS | Thermo Fisher Scientific | Cat #23011 | |
| Chemical compound, drug | Hydroxylamine | Thermo Fisher Scientific | Cat #26103 | |
| Chemical compound, drug | Biotin-HPDP | Soltec Ventures | Cat #B106 | |
| Chemical compound, drug | CNQX | Abcam | Cat #Ab1200017 | |
| Chemical compound, drug | D-AP5 | Abcam | Cat #Ab120003 | |
| Chemical compound, drug | Picrotoxin | Abcam | Cat #Ab120315 | |
| Commercial assay or kit | Pierce BCA Protein assay | Thermo Fisher Scientific | #23225 | |
| Other | High capacity Neutravidin-conjugated beads | Thermo Fisher Scientific | #29202 | |
| Other | Glutathione Sepharose | GE Healthcare | #17075601 | |
| Other | Lipofectamine 2000 | Thermo Fisher Scientific | #11668030 | |
| Software, algorithm | ImageJ Fiji | *Schindelin et al., 2012*; *Schneider et al., 2012* | (RRID:SCR_003070) | |
| Software, algorithm | Clampfit 10 | Molecular Devices | RRID:SCR_011323 | |
| Software, algorithm | Axograph X version 1.6.4 | AxoGraph Scientific | https://axograph.com/ | |
| Software, algorithm | Prism version 8 | GraphPad | RRID:SCR_002798 | |

*Continued on next page*

*Continued*

| Reagent type (species) or resource | Designation | Source or reference | Identifiers | Additional information |
|---|---|---|---|---|
| Cell line (*H. sapiens*) | HEK293T | ATCC | Cat #CRL-3216 (RRID:CVCL_0063) | The identity of the cell line used in this study was authenticated by ATCC using STR profiling and was found to be an exact match to CRL-3216 (HEK293T) in the ATCC STR database. The cells have been tested for mycoplasma nd are negative. |
| Cell line (*H. sapiens*) | AAV-Pro HEK293T | Takara Bio | Cat #632273 | |

## Antibodies

In addition to the commercially available antibodies listed in the Key Resources Table, a ZDHHC14 antibody was raised in rabbits against the peptide CDSLHEDSVRGLVKLSSV (amino acids 473–489 of rat ZDHHC14, plus N-terminal cysteine for conjugation). The resultant antiserum was affinity-purified using the parent immunogenic peptide, and the eluate was then negatively purified using the peptide CLVKLSSV. The resultant flowthrough was dialyzed against PBS and used for experiments.

## Molecular biology and cDNA clones

Full-length mouse *Zdhhc14* cDNA (a kind gift of Dr. M. Fukata) was used as a template to PCR the C-terminal 98 amino acids of ZDHHC14, and the PCR product was subcloned into the yeast expression vector pPC97 for Yeast 2-hybrid experiments. The same PCR product was cloned into pCIS vector 3′ of a GST tag to generate GST fusion protein. The LSSV PDZ ligand was mutated to LSSE by PCR mutagenesis. Mouse *Dlg2* cDNA encoding PSD93α was a gift from Dr. Richard Huganir (Johns Hopkins School of Medicine) and was subcloned into FEWmyc (EF1α promoter) vector to tag with myc on the C-terminus. *Psd93*β cDNA was generated by overhang PCR from *Psd93*α cDNA and the resultant product was subcloned into FEWmyc vector to tag with myc on the C-terminus. PSD93β cysteine mutants were generated either by gene synthesis (Genewiz) or by conventional PCR-based site directed mutagenesis. Full-length *Zdhhc14* cDNA was also cloned into HA-FEW vector downstream of an N-terminal HA epitope tag. LSSE PDZ-mutant *Zdhhc14* cDNA was generated by PCR and was also subcloned into FEW-HA vector. Two shRNA sequences targeting rat *Zdhhc14* (*Zdhhc14*sh#2: GCCACACTCTCAGACATTAT and *Zdhhc14*sh#1 GCATTCAGAGCACCAAATTCGT) were subcloned into FUGW vector (GFP expressing) (*Holland et al., 2016*; *Lois et al., 2002*) together with an H1 promotor to drive shRNA expression. ShRNA efficacy was tested in rat hippocampal neurons.

Vector pAAV-hSyn-hChR2(H134R)-eYFP (Addgene plasmid #26973) was cut with *MluI* and *HindIII* to replace the hSyn-hChR2(H134R)-eYFP fragment with a cassette (synthesized by Genewiz) containing an H1 promoter and shRNA against the kinase DLK, flanked by *PacI* restriction sites (*Holland et al., 2016*) and followed by the human synapsin promoter, an additional PspXI site, Kozak sequence and eGFP cDNA. The resultant vector, termed pAAV-DLKsh-GFP was cut with *PacI* to excise the H1 promoter plus DLK shRNA. The cut vector was then either directly re-ligated to generate pAAV-GFP or ligated with a cassette (synthesized by Genewiz) containing H1 promoter plus *Zdhhc14*sh#1 to generate pAAV-GFP-*Zdhhc14*sh#1.

## Yeast 2-hybrid screen

Two-hybrid screening using the PJ69 yeast strain was performed as previously described (*Thomas et al., 2012*). Briefly, a cDNA coding for the C-terminal 98 amino acids of mouse ZDHHC14 was used to screen a rat hippocampal cDNA library. Clones that grew on quadruple deficient plates (Leu-/Trp-/His-/Ade-) were selected and their plasmids isolated and sequenced. Positive clones were

confirmed by re-transformation of yeast with purified bait and individual 'hit' plasmids and assessing growth on quadruple deficient plates. Back-transformation using a modified prey plasmid coding for the third PDZ domain of PSD93 was performed in HF7C yeast (*Feilotter et al., 1994*).

## HEK293T cell culture and transfection

HEK293T (ATCC, CRL-3216) cells were cultured in Dulbecco's Modified Eagle Medium (DMEM, Thermo Fisher Scientific) supplemented with 10% fetal bovine serum, 1% Penicillin-Streptomycin (Thermo Fisher Scientific), and 1x GlutaMAX Supplement (Thermo Fisher Scientific). A previously described calcium phosphate transfection protocol was used for all HEK293T transfections (*Thomas et al., 2005*). The identity of this cell line was authenticated by ATCC using STR profiling and was found to be an exact match to CRL-3216 (HEK293T) in the ATCC STR database. The cells have been tested for mycoplasma and are negative.

## Hippocampal neuron culture and transfection

Hippocampal neurons were dissociated from dissected embryonic day 18 rat hippocampi as previously described and cultured in Neurobasal with B27 supplement (Thermo Fisher Scientific) (*Thomas et al., 2012*). Lipofectamine 2000 was used to transfect hippocampal neurons on coverslips on day in vitro 12–15 (*Thomas et al., 2012*).

## GST pulldown

Co-precipitation between ZDHHC14 C-terminal tail and PSD93α or β was performed as previously described (*Thomas et al., 2012*). HEK293T cells were cotransfected with pCIS constructs to express GST-ZDHHC14 WT or LSSE C-terminal tails or GST alone and FEW constructs to express C-terminally myc-tagged PSD93α or β. Cells were lysed 16 hr later in immunoprecipitation buffer (IPB: 1x phosphate buffered saline [PBS], 1% [w/v] Triton X-100, 50 mM NaF, 5 mM $Na_4P_2O_7$, 1 mM $Na_3VO_4$, 1 mM EDTA, and 1 mM EGTA plus protease inhibitor cocktail [PIC, Roche, Indianapolis, IN] and microcystin-LR) and insoluble material was removed by centrifugation and passage through a 0.22 µm Spin-X filter column (#8160, Thermo Fisher Scientific). Supernatants were incubated with Glutathione Sepharose (GE Healthcare, Chicago, Il) for 90 min after which beads were washed with IPB, precipitates eluted with 1x SDS sample buffer (SB: 2% SDS [w/v], 50 mM Tris pH 6.8, 10% [v/v] glycerol, 0.005% [w/v] bromophenol blue, and 1% [v/v] β-mercaptoethanol), and subjected to SDS-PAGE followed by western blot as described below.

## Lenti and AAV virus production and shRNA knockdown

VSV-G pseudotyped lentivirus was generated as described (*Thomas et al., 2012*). HEK293T cells were transfected with FUGW vector (with or without *Zdhhc14*sh#1 or #2) vector with untagged *Zdhhc14* cDNA and VSV-G, pMDLg, and RSV-Rev helper plasmids. Lentivirus was concentrated by ultracentrifugation of HEK293T cell media collected 48- and 72 hr post-transfection and resuspended in Neurobasal media (Thermo Fisher Scientific). Infection of hippocampal neurons with control or *Zdhhc14*shRNA virus was performed on DIV9. Neurons were lysed for biochemistry on DIV16 or fixed for immunocytochemistry on DIV16-18.

Adeno-associated viruses were made in AAV-Pro HEK293T cells (Takara Bio) by co-transfecting pAAV-GFP or pAAV-GFP-*Zdhhc14*sh#1 with pAAV2 (pACG2)-RC triple mutant (Y444, 500, 730F) (*Petrs-Silva et al., 2011*) and pHelper (Stratagene, La Jolla, California) plasmids. Cells were lysed 72 hr post-transfection to release viral particles, which were precipitated using 40% (w/v) polyethylene glycol and purified by cesium chloride density gradient centrifugation. Fractions with refractive index from 1.370 to 1.374 were dialyzed in MWCO 7000 Slide-A-Lyzer cassettes (Thermo Fisher Scientific, Waltham, Massachusetts) overnight at 4°C. AAV titers used for this study were in the range of 1.5–$2.5 \times 10^{12}$ genome copies (GC)/ml determined by real-time PCR.

## Acyl-biotin exchange assay (ABE)

Palmitoylation levels were determined as previously described (*Thomas et al., 2012*). 8 hr after transfection HEK293T cells were lysed and sonicated in ABE lysis buffer (50 mM HEPES pH 7.0, 2% [w/v] SDS, 1 mM EDTA plus PIC) with 20 mM thiol-reactive methyl-methane thiosulfonate (MMTS, Thermo Fisher Scientific) to block free cysteine residues. Transduced hippocampal neurons were

lysed and sonicated in the same lysis buffer on DIV16 and lysates passed through a QIAshredder homogenization column (#79654, Qiagen, Hilden, Germany). HEK293T cell and hippocampal neuron lysates were then incubated at 50°C for 20 min after which protein was precipitated and excess MMTS removed by acetone precipitation. Protein pellets were then dissolved in 4% SDS buffer (4% [w/v] SDS, 50 mM Tris pH 7.5, 5 mM EDTA plus PIC) and the resultant solution wassplit in two and incubated for one hour with either 0.7 M hydroxylamine pH 7.4 (+NH$_2$OH, HAM, Thermo Fisher Scientific), to cleave thioester bonds and remove palmitate groups, or with 50 mM Tris pH 7.4 (buffer control, -NH$_2$OH). Both buffers contained 1 mM biotin-HPDP (Soltec Ventures, Beverly, MA) to biotinylate newly revealed cysteines. Protein was then acetone precipitated again to remove excess hydroxylamine and biotin-HPDP and pellets were re-dissolved in lysis buffer plus PIC without MMTS and then diluted 1:20 in dilution buffer (50 mM HEPES, 1% [v/v] Triton X-100, 1 mM EDTA, 1 mM EGTA, 150 mM NaCl plus protease inhibitors). Biotinylated proteins were then purified using high capacity neutravidin-conjugated beads (Thermo Fisher Scientific) via incubation for three hours at 4°C after which beads were washed extensively in dilution buffer plus 0.5M NaCl. Purified protein was then eluted by HPDP reduction with 1% (v/v) β-mercaptoethanol (Millipore-Sigma) in elution buffer (0.2% [w/v] SDS and 0.25M NaCl in dilution buffer) for 10 min at 37°C. Eluted protein was then denatured in SB and subjected to SDS-PAGE followed by western blot as described below.

## Western blot and quantification

For developmental time course experiments hippocampal neurons were lysed and sonicated in ABE lysis buffer with PIC on the indicated DIV. Lysates were passed through a 0.22 µm Spin-X filter column after which total protein amount was determined using the Pierce BCA Protein Assay Kit (#23225, Thermo Fisher Scientific). Equal protein amounts were denatured in SB and subjected to SDS-PAGE. SDS-PAGE gels with GST precipitates, ABE samples, or developmental time course samples were transferred to Immobilon-P PVDF (#IPVH00010, 0.45 µm, Millipore-Sigma) membranes, blocked in 5% (w/v) skim milk/Tris buffered saline (TBS), and immunoblotted with indicated antibodies. HRP conjugated secondary antibodies were used for ECL-mediated visualization (Western Lightening Plus-ECL, #NEL105001EA, Perkin Elmer, Waltham, MA) with film (GeneMate Blue Lite Autorad Film, F-#9024−8 × 10, VWR, Radnor, PA). Independent neuronal cultures or HEK293T transfections from separate passages were considered biological replicates. In both cases individual plates of cells were randomly assigned to the various groups. Uncropped western blot images are shown in the following figure supplements corresponding to the parent figure: *Figure 1—figure supplement 4*, *Figure 2—figure supplement 2*, *Figure 4—figure supplement 2*, *Figure 5—figure supplement 2*, and *Figure 6—figure supplement 1*. Quantifications were performed using Image Studio Lite.

## Immunostaining

To determine AIS targeting of endogenous proteins, transduced neurons on coverslips were fixed in 2% (w/v) paraformaldehyde (PFA, Electron Microscopy Sciences, Hatfield, PA) in 1xPBS with 4% (w/v) sucrose for 15 min at 4°C. Transfected neurons were fixed in 4% (w/v) PFA in 1xPBS with 4% (w/v) sucrose for 10 min at room temperature. All coverslips were then washed three times (five minutes per wash) in 1xPBS followed by block and permeabilization with 5% (v/v) normal goat serum (NGS, Southern Biotech) and 0.25% (v/v) Triton X-100 in 1xPBS for 1 hr at room temperature. Coverslips were incubated with indicated primary antibodies overnight at room temperature in permeabilization/block solution, washed with PBS and then incubated with Alexa-Fluor conjugated secondary antibodies for one hour at room temperature before mounting on glass slides with FluorSave Reagent (Millipore-Sigma, #345789). Specificity of anti-PSD93, anti-Kv1.1, anti-Kv1.2, anti-Kv1.4, and anti-ankyrin-G antibodies in immunocytochemical studies was verified by control experiments omitting the primary antibody in question, which, in each case, abolished staining.

## Image acquisition and analysis

Immunostained neurons were imaged using a Nikon C2 inverted confocal microscope with a 60x oil immersion objective (1.4 NA, plan-Apo). Z-stack images were taken with 0.4 µm spacing at 1024 × 1024 resolution and parameters were kept constant between different conditions. In all experiments wells of neurons were randomly assigned to the various groups.

For experiments involving AIS protein targeting a field of interest was selected based on the AIS marker (AnkG) with no knowledge of the experimental signal (PSD93, Kv1.1–1.4) that was subsequently acquired in a separate channel. Confocal image stacks were opened using ImageJ Fiji software (*Schindelin et al., 2012*; *Schneider et al., 2012*) and background subtraction was performed using a rolling ball radius of 50 pixels. AIS targeting was determined using the 3D Objects Counter plugin (*Bolte and Cordelières, 2006*) in ImageJ Fiji by thresholding the AnkG signal to generate a 3-dimensionally reconstructed mask of regions of interest corresponding to individual AISs. The mask was then used to quantify mean intensity and integrated density within each AIS in the corresponding PSD93 or Kv1 channel image (again in three dimensions) using the 'Redirect to' function in the 3D Object Counter plugin. Thresholding settings were maintained through an individual experiment image set and a lower limit size filter of 300 pixels was set to exclude non-AIS 3D objects. Data from particles not corresponding to intact AISs were manually removed prior to analysis. In each individual experimental replicate, i.e. an independent culture, a minimum of 15, and a mean of 85 individual AISs were quantified and averages were calculated to generate a single determination for both mean intensity and integrated density that was considered a biological replicate.

AIS lengths were determined by tracing thresholded AnkG images through the confocal stack to generate a 3D tracing of each individual AIS using the using the Simple Neurite Tracer plugin (*Longair et al., 2011*) in ImageJ Fiji. Thresholding settings were maintained through an individual experiment image set. In each individual experimental replicate, defined as an independent culture, at least 50–150 individual AISs were traced and an average AIS length was calculated to generate a single determination that was considered a biological replicate.

Images of spine-bearing (i.e. presumptive excitatory) PSD93β-myc transfected neurons were acquired and an experimenter blind to condition scored AIS myc distribution manually using maximum intensity projections generated in ImageJ Fiji. Ten transfected neurons were imaged and scored per condition in each individual transfected culture and the percentage value of AIS-enriched versus diffuse signal per culture was considered a biological replicate.

For illustration purposes images used in figures are maximum intensity projections generated and modified for brightness and contrast using ImageJ Fiji where settings were maintained across each example set of images.

## Electrophysiological recordings

For electrophysiological experiments, neurons were cultured as described (*Moon et al., 2013*) and infected at DIV3 with AAVs. At 13–15 days post-infection whole-cell patch-clamp recordings were performed at room temperature. Cells were briefly viewed for EGFP fluorescence to confirm infection (Olympus FITC/GFP filter cube). Whole-cell recordings were obtained using borosilicate glass electrodes (resistance 3–5 MΩ) filled with intracellular solution containing the following (in mM): 130 potassium methylsulfate, 10 KCl, 5 Tris-phosphocreatine, 10 HEPES, 4 NaCl, 4 $Mg_2ATP$, and 0.4 $Na_4GTP$. The pH was adjusted to 7.2 to 7.3 with KOH. The extracellular solution contained the following (in mM): 125 NaCl, 26 $NaHCO_3$, 2.5 KCl, 1 $NaH_2PO_4$, 1.3 $MgCl_2$, 2.5 $CaCl_2$, and 12 glucose, saturated with 95% $O_2$ and 5% $CO_2$. NBQX (4 µM), D-AP5 (10 µM), and picrotoxin (100 µM; Abcam) were included in the extracellular medium to block AMPA-, NMDA-, and GABA-mediated synaptic transmission, respectively. For all current-clamp experiments membrane potential was kept between −60 and −65 mV by applying small DC current. For the voltage-clamp experiments we used the same intracellular solution as for the current-clamp recordings. Neurons were held at −70 mV and a series of voltage steps were given from −80 mV to +100 mV in increments of 15 mV. Data were acquired using Multiclamp 700B amplifier (Molecular Devices) and digitized using Digidata 1440A (Molecular Devices). All data were low-pass filtered at 10 kHz and sampled at 50 kHz. Individual neurons were considered biological replicates and wells of neurons were randomly assigned to the various groups. Data were analyzed off-line using Clampfit (Molecular Devices).

## Statistical analysis

All data were analyzed using GraphPad Prism eight software (GraphPad Software, San Diego, CA). In all graphs the mean is shown and error bars indicate standard deviation of the mean (SD). Shapiro-Wilk normality test was used to determine if data were normally distributed. Statistical test used is indicated in the corresponding figure legend or table column. In cases with two experimental

groups, if data were normally distributed unpaired Student's t-test was used but if data were not normally distributed or variances between groups were unequal (determined by F test) a non-parametric Mann Whitney t-test was used. In cases with more than two experimental groups, all data were normally distributed, thus 1-way ANOVA was used unless variances between groups were unequal (determined by Bartlett's test), in which case Welch's 1-way ANOVA was used instead. In cases with more than one independent variable without repeated measures, two-way ANOVA was used. When repeated electrophysiological measurements were taken on individual neurons with increasing current steps, neurons fired 0-X number of action potentials and the resulting data were analyzed using a repeated measures 2-way ANOVA. However, when a neuron did not fire any action potentials at a given current step there was no time value for latency to first action potential resulting in missing values so instead a mixed effects analysis was used.

## Acknowledgements

We thank Drs. Dale Martin (University of Waterloo, ON, Canada), Jingwen Niu, and Heykyeong Jeong (both Thomas lab, Temple University) for invaluable input. We thank Drs. Fengsong Qin and George Smith of the Shriners Hospitals Pediatric Research Center Viral Production Core (Temple University) for AAV production, Dr. Richard Huganir (Johns Hopkins University School of Medicine) for providing hippocampal cDNA library and PSD93α cDNA, Dr. Maskai Fukata (National Institute of Physiological Sciences, Okazaki, Japan) for *Zdhhc14* cDNA, and Drs. Paul Jenkins (University of Michigan) and Christophe Leterrier (Aix Marseille Université, France) for expertise and advice in image analysis. This work was supported by NIH (grant R01NS094402), seed funding and unrestricted funds from Shriners Hospitals for Children and Temple University (all to GT), and by a Brody Family Medical Trust Fund Fellowship (to SS). AAV production was supported by a Special Shared Facilities grant from Shriners Hospitals for Children to Dr. George M Smith.

## Additional information

### Funding

| Funder | Grant reference number | Author |
|---|---|---|
| Brody Family Medical Trust Fund | Postdoctoral Fellowship | Shaun S Sanders |
| NIH | R01NS094402 | Gareth M Thomas |
| Shriners Hospitals for Children | Seed funding | Gareth M Thomas |
| Temple University | Seed funding | Gareth M Thomas |

The funders had no role in study design, data collection and interpretation, or the decision to submit the work for publication.

### Author contributions

Shaun S Sanders, Conceptualization, Data curation, Formal analysis, Investigation, Writing - original draft, Writing - review and editing; Luiselys M Hernandez, Conceptualization, Data curation, Formal analysis, Investigation, Writing - review and editing; Heun Soh, Data curation, Formal analysis, Investigation; Santi Karnam, Data curation, Investigation; Randall S Walikonis, Investigation, Methodology, Writing - review and editing; Anastasios V Tzingounis, Data curation, Formal analysis, Supervision, Investigation, Methodology, Writing - original draft, Writing - review and editing; Gareth M Thomas, Conceptualization, Data curation, Formal analysis, Supervision, Funding acquisition, Investigation, Methodology, Writing - original draft, Project administration, Writing - review and editing

### Author ORCIDs

Shaun S Sanders https://orcid.org/0000-0001-9661-141X
Anastasios V Tzingounis http://orcid.org/0000-0002-4605-3437
Gareth M Thomas https://orcid.org/0000-0003-3183-8431

## Ethics

Animal experimentation: This study was performed in strict accordance with the recommendations in the Guide for the Care and Use of Laboratory Animals of the National Institutes of Health. All animals were handled according to approved institutional animal care and use committee (IACUC) protocols (#4939) of Temple University.

## Decision letter and Author response

Decision letter https://doi.org/10.7554/eLife.56058.sa1
Author response https://doi.org/10.7554/eLife.56058.sa2

## Additional files

### Supplementary files

• Transparent reporting form

### Data availability

All data generated during this study are included in the manuscript and supporting files. Source data files have been provided for all figures in the source data excel file.

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
