## [Decision Letter]

**Acceptance summary:**

Macromolecular complexes in neurons are play important roles in the physiology of neurons, however, how ion channels and other proteins are localized to the initial segment of the axon is not well understood. In this manuscript the authors examine the role of the palmitoyl acyltransferase (PAT) ZDHHC14 in regulating clustering of Kv1 channels via PSD93 at the axon initial segment (AIS). The novel observations in this study are that ZDHHC14 drive palmitoylation of PSD93, which promotes the clustering of PSD93 to the AIS, providing a scaffold to facilitate targeting and clustering of Kv1 channels to the AIS. The functional relevance of these observations is underscored by data suggesting that the down-regulation of ZDHHC14 increases the excitability of hippocampal neurons.

**Decision letter after peer review:**

Thank you for submitting your article "The palmitoyl acyltransferase ZDHHC14 controls ion channel clustering at the axon initial segment" for consideration by *eLife*. Your article has been reviewed by three peer reviewers, including Leon D Islas as the Reviewing Editor and Reviewer #1, and the evaluation has been overseen by John Huguenard as the Senior Editor. The following individual involved in review of your submission has agreed to reveal their identity: Maren Engelhardt (Reviewer #3).

The reviewers have discussed the reviews with one another and the Reviewing Editor has drafted this decision to help you prepare a revised submission.

As the editors have judged that your manuscript is of interest, but as described below that additional experiments are required before it is published, we would like to draw your attention to changes in our revision policy that we have made in response to COVID-19 (https://elifesciences.org/articles/57162). First, because many researchers have temporarily lost access to the labs, we will give authors as much time as they need to submit revised manuscripts. We are also offering, if you choose, to post the manuscript to bioRxiv (if it is not already there) along with this decision letter and a formal designation that the manuscript is “in revision at *eLife*”. Please let us know if you would like to pursue this option. (If your work is more suitable for medRxiv, you will need to post the preprint yourself, as the mechanisms for us to do so are still in development.)

Summary:

The mechanisms by which macromolecular complexes are integrated in neurons are relatively well known when it comes to synapse formation, however how ion channels and other proteins are localized to the initial segment of the axon is not well understood. The goal of this manuscript was to examine the role of the palmitoyl acyltransferase (PAT) ZDHHC14 in regulating the clustering of Kv1 channels via PSD93 at the axon initial segment (AIS). The novel observations in this study are that ZDHHC14 drive palmitoylation of PSD93. This post-translational modification promotes the clustering of PSD93 to the AIS. In doing so, PSD93 seems to serve as a scaffold to facilitate targeting and clustering of Kv1 channels to the AIS. The relevance of these observations is underscored by data suggesting that the downregulation of ZDHHC14 increases the excitability of hippocampal neurons. In general, the paper is well written, uses well-established approaches in the field, and the data seem of high quality, with appropriate controls and in support of the main hypothesis.

Essential revisions:

1) Statistical analyses are insufficient and in their current form overstate the data. For example, the authors fail to explain what type of data distribution is expected and which is the appropriate chosen test (figure legends indicate all tests used, but it remains a mystery to understand the abundant use of t-tests for example). When using descriptive statistics (means), the authors incorrectly show S.E.M. instead of S.D. The sample size (N) varies quite dramatically sometimes within the same graph and analysis – it is unclear how sometimes highly significant results are obtained from comparisons of N=3-13 within the same test.

2) The authors chose to treat individual neurons (Figure 3 and supplements, Figure 4, Figure 7) as biological replicates. This is a problematic decision for several reasons. AIS show significant heterogeneity in length, onset and location as shown by numerous recent studies (Thome et al., 2014 Neuron, Hamada et al., 2016 PNAS, Hoefflin et al., 2017 Front Cell Neurosci, Meza et al., 2018 J Neurosci). This has direct impact on neuronal excitability (Gulledge and Bravo, 2015 eNeuro, Kole and Brette, 2018 Curr Opin Neurosci, Goethals and Brette, 2020 *eLife*). Therefore, individual neurons vary quite a bit in all of these parameters, including possibly location and distribution of putative binding sites. Therefore when statements in the Discussion are based on these data, an issue arises (for example Discussion, fourth and fifth paragraphs). It is far more accurate to treat individual cultures as N, with at least 50-100 AIS/culture. This is what the vast majority of current papers on the AIS field reflect. The authors do not have to perform new experiments, but rather compare their existing data differently. It would give a far more accurate read-out.

3) The image analysis for AIS measurements of expression as described in the Materials and methods section is inadequate to draw meaningful conclusions about clustering. Co-localization analysis in confocal stacks requires 3D reconstruction and not just a change in "color" of two overlaying channels in Maximum Intensity Projections. The same issue applies to AIS length measurements.

4) The conclusions drawn based on data in Figures 5 and 8 are overstated. Immunoblot analysis of Kv1.x expression from entire neuron lysates takes into consideration all Kv expression – somatodendritic as well as axonal/AIS. It is by no means an indicator of a developmental expression at the AIS (see Discussion). Such an experiment would require either the laser-dissection of individual AIS in sufficient numbers to run immunoblots, or a comprehensive, semi-quantitative IF analysis at individual AIS. These experiments are not essential, but the authors should tone down the conclusions and also state that all Kvs will be detected in an immunoblot assay, regardless of their subcellular localization. Figure 8: In the Results, this data is used to suggest that: "ZDHHC14 predominantly controls palmitoylation of PSD93 and Kv1 within the Golgi". However, the authors provide no quantification of their IF data.

5) Electrophysiology. While potentially very informative, the data presented are not sufficient to conclude that it is indeed the loss of ZDHCC14 and therefore Kv channels at the AIS that result in an increase in excitation. The authors should report at least all passive properties (table format) across samples. What about current threshold, threshold voltage, half-width, and an analysis of AP waveform? As they stand, electrophysiological experiments inform of a possible physiological role played by palmitoylated kv1 channels, however, the reduced threshold could also be produced by down-regulation of other outward potassium conductances or increases of excitatory inward currents as a direct consequence of ZDHHC14 knockdown. The authors should provide direct evidence (by voltage-clamp) that the neurons with increased excitability show a reduced outward current that can be ascribed to kv1 channels. These should be easy experiments to perform.

6) All raw Western blot films should be provided as supplementary data.

[Editors' note: further revisions were suggested prior to acceptance, as described below.]

Thank you for resubmitting your article "The Palmitoyl Acyltransferase ZDHHC14 Controls Kv1-Family Potassium Channel Clustering at the Axon Initial Segment" for consideration by *eLife*. Your revised article has been overseen by Leon Islas as Reviewing Editor and John Huguenard as the Senior Editor.

The Reviewing Editor has drafted this decision to help you prepare a revised submission.

The authors have made a great effort of addressing the comments of reviewers, including new experiments and greatly improving the statistical analysis as well as analysis if imaging data. The new recordings of potassium currents greatly support the conclusions drawn from electrophysiology. However, currents should be presented as current density (current magnitude/ cell capacitance) to be correctly interpreted as a reduction in channels density or channel expression. Please attend this correction before a final decision on the manuscript.

---

## [Author Response]

Essential revisions:1) Statistical analyses are insufficient and in their current form overstate the data. For example, the authors fail to explain what type of data distribution is expected and which is the appropriate chosen test (figure legends indicate all tests used, but it remains a mystery to understand the abundant use of t-tests for example). When using descriptive statistics (means), the authors incorrectly show S.E.M. instead of S.D. The sample size (N) varies quite dramatically sometimes within the same graph and analysis – it is unclear how sometimes highly significant results are obtained from comparisons of N=3-13 within the same test.

We appreciate the reviewers raising this point. Data in our revised manuscript are now re-plotted using SD rather than SEM. In addition, to better represent the experimental variation in the data, we now plot all data normalized to control across biological replicates rather than normalized to the control in each individual experiment. We also performed normality tests and F or Bartlett’s tests of equal variation of all data sets and now only use t-tests when the data are normally distributed and groups have equal variation. If the data do not pass these two tests, we instead perform a non-parametric Mann Whitney test. We explain these analyses in more detail the revised Materials and methods section “Statistical analyses”.

We also appreciate the reviewers’ concern with the varying N in some individual experiments. In the example they highlight, this issue occurred because we first assessed palmitoylation of only one PSD93 isoform by wild type ZDHHC14 and only later assessed palmitoylation of a second PSD93 isoform, and the effect of the ZDHHC14-LSSE mutation. Because a wtZDHHC14 and PSD93 “reference” sample was always included in the later experiments we pooled all data, but the N number for the “wt ZDHHC14 and PSD93” condition was hence disproportionately greater. We have now removed experiments that did not include a ZDHHC14-LSSE condition and analyzed the data in two ways. In a revised main Figure 1 we present the data with the alpha and beta subunits analyzed separately where the N are equal across all conditions (N=6 and N=4, respectively) and in a new Figure 1—figure supplement 3 we show the data plotted and analyzed together where there is a much smaller discrepancy between N. The N in these updated statistical analyses are thus much more similar.

We thank the reviewers for bringing up these points and hope they would agree that our updated analyses better represent the data. Importantly, all of the major effects that we initially reported remain statistically significant with these revised tests and methods of analysis.

2) The authors chose to treat individual neurons (Figure 3 and supplements, Figure 4, Figure 7) as biological replicates. This is a problematic decision for several reasons. AIS show significant heterogeneity in length, onset and location as shown by numerous recent studies (Thome et al., 2014 Neuron, Hamada et al., 2016 PNAS, Hoefflin et al., 2017 Front Cell Neurosci, Meza et al., 2018 J Neurosci). This has direct impact on neuronal excitability (Gulledge and Bravo, 2015 eNeuro, Kole and Brette, 2018 Curr Opin Neurosci, Goethals and Brette, 2020 eLife). Therefore, individual neurons vary quite a bit in all of these parameters, including possibly location and distribution of putative binding sites. Therefore when statements in the Discussion are based on these data, an issue arises (for example Discussion, fourth and fifth paragraphs). It is far more accurate to treat individual cultures as N, with at least 50-100 AIS/culture. This is what the vast majority of current papers on the AIS field reflect. The authors do not have to perform new experiments, but rather compare their existing data differently. It would give a far more accurate read-out.

We appreciate this point from the reviewers. As suggested, we revised our analyses using individual cultures (rather than AIS’s) as N and also used 3D reconstruction (see Point 3 below) rather than maximum intensity projections. Under these new analysis conditions, we found that the effect of ZDHHC14 loss on AIS localization of PSD93, Kv1.1, Kv1.2, and Kv1.4 remained statistically significant, as did the change in AIS length. We have re-plotted the data to reflect these changes and modified the Results and Discussion text to describe these modified analyses and our conclusions. As with point #1, we note that the major effects that we initially reported largely remain statistically significant with these revised analyses. For Kv1.2, the mean gray value data do not reach significance with this modified analysis, but the integrated density data remain significant i.e. the total amount of Kv1.2 at the AIS is still reduced (Figure 3, Figure 3—figure supplement 3, and Figure 7).

3) The image analysis for AIS measurements of expression as described in the Materials and methods section is inadequate to draw meaningful conclusions about clustering. Co-localization analysis in confocal stacks requires 3D reconstruction and not just a change in "color" of two overlaying channels in Maximum Intensity Projections. The same issue applies to AIS length measurements.

We thank the reviewers for raising this important point. As suggested, we reanalyzed both AIS targeting and AIS lengths using 3D reconstruction. As explained in our response to Point 2, our five readouts (PSD93 targeting, Kv1.1 targeting, Kv1.2 targeting, Kv1.4 targeting, and AIS length) remained statistically significant when 3D reconstructions were used and when data were analyzed using individual cultures as N (Points 3 and 2, respectively). We have modified our Materials and methods (“Image acquisition and analysis” section) and figures to reflect these new analyses (Figure 3, Figure 3—figure supplement 3, and Figure 7). We have also updated our Materials and methods section to fully explain these updated analysis procedures. For clarification, we also note that although the initial Point #3 includes the term “co-localization”, our AIS targeting analyses do not assess co-localization per se – instead the AnkG signal is used as a mask and the PSD93/Kv1.X signal within the masked region is then calculated. Importantly, though, the reviewers’ suggestion to use a mask generated from a 3D reconstruction of the entire stack of images, rather than a 2-dimensional Maximum Intensity projection greatly reduced contributions from non-AIS-localized (e.g. synaptodendritic) PSD93/Kv1.X signals from other neurons in the culture. We are hence very grateful to the reviewers for suggesting this modified analysis, as we hope they would agree that it far more accurately captures the effects of interest.

4) The conclusions drawn based on data in Figures 5 and 8 are overstated. Immunoblot analysis of Kv1.x expression from entire neuron lysates takes into consideration all Kv expression – somatodendritic as well as axonal/AIS. It is by no means an indicator of a developmental expression at the AIS (see Discussion). Such an experiment would require either the laser-dissection of individual AIS in sufficient numbers to run immunoblots, or a comprehensive, semi-quantitative IF analysis at individual AIS. These experiments are not essential, but the authors should tone down the conclusions and also state that all Kvs will be detected in an immunoblot assay, regardless of their subcellular localization. Figure 8: In the Results, this data is used to suggest that: "ZDHHC14 predominantly controls palmitoylation of PSD93 and Kv1 within the Golgi". However, the authors provide no quantification of their IF data.

These are important points and we apologize for the initial overstatement. We have therefore toned down the conclusions of our developmental expression analysis as suggested. We have also softened the language regarding possible location(s) in which ZDHHC14-dependent palmitoylation of PSD93 and Kv1 occurs. However, we do include an additional reference (Sanchez-Ponce et al., 2012) that shows (albeit non-quantitatively) that Kv1.2 is first detected at the AIS at a very similar timepoint (DIV13) to when we observe marked upregulation of Kv1.2 protein levels (DIV12) in similar hippocampal cultures. We briefly mention this point in our revised Discussion, while also highlighting the caveats raised by reviewers (second paragraph of the “Developmental expression of ZDHHC14 mirrors that of PSD93 and Kv1 channels” Results section, the “ZDHHC14 is a major neuronal PAT for Kv1.1, Kv1.2, and Kv1.4” Results section, the “ZDHHC14 is predominantly a Golgi-localized PAT in hippocampal neurons” Results section, and the third and fifth paragraph of the Discussion).

5) Electrophysiology. While potentially very informative, the data presented are not sufficient to conclude that it is indeed the loss of ZDHCC14 and therefore Kv channels at the AIS that result in an increase in excitation. The authors should report at least all passive properties (table format) across samples. What about current threshold, threshold voltage, half-width, and an analysis of AP waveform? As they stand, electrophysiological experiments inform of a possible physiological role played by palmitoylated kv1 channels, however, the reduced threshold could also be produced by down-regulation of other outward potassium conductances or increases of excitatory inward currents as a direct consequence of ZDHHC14 knockdown. The authors should provide direct evidence (by voltage-clamp) that the neurons with increased excitability show a reduced outward current that can be ascribed to kv1 channels. These should be easy experiments to perform.

We thank the reviewers for encouraging us to perform additional analyses and experiments to address this point. Indeed, we found that key features of the action potential properties in Zdhhc14 knockdown neurons are consistent with reduced Kv1 channel number and/or function. In particular, closer examination of our data revealed that the latency to the first action potential, rheobase, and the rate of action potential repolarization were all decreased in Zdhhc14 knockdown neurons (Figure 9D-G; Table 1). These data are consistent with reduced Kv1 levels and/or function at the AIS because Kv1-type channels are known to regulate action potential firing by suppressing generation of, as well as shortening, action potentials (Yamada and Kuba, 2016). In addition, the slight but significant AP broadening in Zdhhc14 knockdown neurons (Table 1) is consistent with that seen following acute block of Kv1 channels (Guan, Pathak, and Foehring, 2018; Kole, Letzkus and Stuart, 2007). Moreover, our new experiments showing markedly reduced outward currents in the absence of Zdhhc14 further support the notion that the increased excitability is due to loss of Kv1 channels (Figure 9A-C). Together, these new data and analyses are all consistent with our model in which ZDHHC14 regulates Kv1 localization and/or function. Despite this additional supportive evidence, we are nonetheless careful to note that additional ZDHHC14 substrate(s) may contribute to these effects. We have updated Figure 9 to include these new data and have also added a new Table 1 to summarize the results of the action potential analyses. In addition, we briefly expanded our Discussion to highlight how these new data support our overall model (seventh and tenth paragraphs of the Discussion).

6) All raw Western blot films should be provided as supplementary data.

Images of all raw Western blot films are now provided in additional figure supplements relating to the corresponding parent figure, as requested.

[Editors' note: further revisions were suggested prior to acceptance, as described below.]

The authors have made a great effort of addressing the comments of reviewers, including new experiments and greatly improving the statistical analysis as well as analysis if imaging data. The new recordings of potassium currents greatly support the conclusions drawn from electrophysiology. However, currents should be presented as current density (current magnitude/ cell capacitance) to be correctly interpreted as a reduction in channels density or channel expression. Please attend this correction before a final decision on the manuscript.

We appreciate this point from the reviewers. Figure 9C is now re-plotted to reflect this change and the accompanying text of the Figure 9 legend, including the statistical analysis, has been updated accordingly.